# The role of photorespiration during the evolution of C$_4$ photosynthesis in the genus *Flaveria*

**Julia Mallmann[1†], David Heckmann[2†], Andrea Bräutigam[3], Martin J Lercher[2,4], Andreas PM Weber[3,4], Peter Westhoff[1,4], Udo Gowik[1*]**

[1]Institute for Plant Molecular and Developmental Biology, Heinrich-Heine-Universität, Düsseldorf, Germany; [2]Institute for Computer Science, Heinrich-Heine-Universität, Düsseldorf, Germany; [3]Institute of Plant Biochemistry, Heinrich-Heine-Universität, Düsseldorf, Germany; [4]Cluster of Excellence on Plant Sciences (CEPLAS), Heinrich-Heine-Universität, Düsseldorf, Germany

**Abstract** C$_4$ photosynthesis represents a most remarkable case of convergent evolution of a complex trait, which includes the reprogramming of the expression patterns of thousands of genes. Anatomical, physiological, and phylogenetic and analyses as well as computational modeling indicate that the establishment of a photorespiratory carbon pump (termed C$_2$ photosynthesis) is a prerequisite for the evolution of C$_4$. However, a mechanistic model explaining the tight connection between the evolution of C$_4$ and C$_2$ photosynthesis is currently lacking. Here we address this question through comparative transcriptomic and biochemical analyses of closely related C$_3$, C$_3$–C$_4$, and C$_4$ species, combined with Flux Balance Analysis constrained through a mechanistic model of carbon fixation. We show that C$_2$ photosynthesis creates a misbalance in nitrogen metabolism between bundle sheath and mesophyll cells. Rebalancing nitrogen metabolism requires anaplerotic reactions that resemble at least parts of a basic C$_4$ cycle. Our findings thus show how C$_2$ photosynthesis represents a pre-adaptation for the C$_4$ system, where the evolution of the C$_2$ system establishes important C$_4$ components as a side effect.

*For correspondence: gowik@ uni-duesseldorf.de

†These authors contributed equally to this work

**Competing interests:** The authors declare that no competing interests exist.

**Reviewing editor**: Detlef Weigel, Max Planck Institute for Developmental Biology, Germany

## Introduction

The dual-specific enzyme ribulose 1,5-bisphosphate carboxylase/oxygenase (Rubisco) catalyzes two opposing reactions—the carboxylation and the oxygenation of ribulose 1,5-bisphosphate. The former reaction yields 3-phosphoglycerate (3-PGA), whereas the latter produces 2-phosphoglycolate (2-PG). 3-PGA is reduced to carbohydrates in the Calvin–Benson cycle and incorporated into biomass. However, 2-PG is toxic, which requires its removal by a metabolic repair pathway called photorespiration (*Anderson, 1971*; *Bowes et al., 1971*; *Ogren, 1984*; *Leegood et al., 1995*). In the photorespiratory cycle, 2-PG is regenerated to 3-PGA, but it involves the release of formerly assimilated CO$_2$ and NH$_3$, entails energy costs for the plants and reduces the efficiency of photosynthesis by up to 30% (*Ehleringer et al., 1991*; *Bauwe et al., 2010*; *Raines, 2011*; *Fernie et al., 2013*). Eight core enzymes are required for photorespiration, which in higher plants are located in the chloroplast, the peroxisome, and the mitochondrion (*Bauwe et al., 2010*; *Figure 1A*). The pathway rescues ¾ of the carbon, which would otherwise be lost through the oxygenase activity of Rubisco (*Peterhansel et al., 2010*; *Fernie et al., 2013*). Ammonia refixation in the chloroplast by the combined activities of glutamine synthase (GS) and glutamine oxoglutarate aminotransferase (GOGAT) is an integral part of photorespiration.

In hot and dry environments and under low atmospheric CO$_2$ conditions, when the oxygenation activity of Rubisco is increased, the high rate of photorespiration becomes unfavorable for the plants

**eLife digest** Environmental pressures sometimes cause different organisms to independently evolve the same traits. A dramatic example of this phenomenon, which is called convergent evolution, can be seen in the modes used by plants to convert carbon dioxide from the air into starch during photosynthesis.

Early plants existed in an environment with high levels of carbon dioxide in the air. Over time, carbon dioxide levels decreased, so plants evolved more efficient types of photosynthesis to cope. A very efficient type of photosynthesis, called $C_4$ photosynthesis essentially represents a carbon dioxide concentration mechanism. It has evolved at least 62 times independently in 19 different families of flowering plants.

Scientists have shown that a less advanced, low-efficiency version of photosynthetic carbon dioxide concentration, called $C_2$ photosynthesis, is a stepping-stone to $C_4$ photosynthesis. It is also known that the evolution of $C_4$ photosynthesis required changes to the expression patterns of thousands of genes, but the exact mechanism that leads from $C_2$ photosynthesis to $C_4$ photosynthesis is not clear.

To explore this in greater detail, Mallmann, Heckmann et al. studied plants from the genus *Flaveria*, which belongs to the same family as sunflowers and asters. Under identical greenhouse conditions, plants that use three different photosynthetic pathways—$C_3$ photosynthesis, $C_4$ photosynthesis, or an intermediate between the two—were grown and their gene expression patterns were compared. Computer simulations were used to model the metabolism of plants that relied on $C_2$ photosynthesis.

Based on the modeling, it appears that $C_2$ photosynthesis shifts the balance of nitrogen metabolism between two types of cell that are critical to photosynthesis. To rebalance the nitrogen, several genes are expressed to trigger an ammonia recycling mechanism. The same genes are turned on during $C_4$ photosynthesis, and this recycling mechanism include parts of the $C_4$ process.

The findings of Mallmann, Heckmann et al. suggest that the initial steps in $C_4$ photosynthesis evolved to prevent nitrogen imbalance. Over time, this mechanism was co-opted to become part of a more efficient form of photosynthesis, which may explain why so many different plants evolved from $C_2$ to $C_4$ photosynthesis.

(*Sage, 2001*, *2013*). $C_4$ plants possess a mechanism that minimizes the oxygenase function of Rubisco and thereby reduces photorespiration and decreases the loss of carbon. $C_4$ photosynthesis is based on a division of labor between two different cell types, mesophyll and bundle sheath cells, which are organized in a wreath-like structure called 'Kranz Anatomy' (*Haberlandt, 1904*; *Dengler and Nelson, 1999*). Atmospheric $CO_2$ is initially fixed in the mesophyll by phosphoenolpyruvate carboxylase (PEPC), and the resulting four-carbon compound is transported to the bundle sheath cells and decarboxylated by NADP/NAD malic enzyme or phosphoenolpyruvate carboxykinase (*Hatch et al., 1975*). Thereby $CO_2$ is concentrated at the site of the Rubisco in the bundle sheath cells (*Hatch, 1987*), outcompeting the molecular oxygen. As a consequence, photorespiration is drastically reduced as compared to $C_3$ plants, and $C_4$ plants are characterized by a high photosynthetic efficiency (*Figure 1B*).

$C_4$ plants have evolved multiple times independently from $C_3$ ancestors. The evolution of $C_4$ photosynthesis occurred at least 62 times in 19 different families of the angiosperms (*Sage et al., 2011*), implying a low evolutionary barrier towards expression of this trait. The analysis of recent intermediate species (*Bauwe and Kolukisaoglu, 2003*; *Sage, 2004*; *Bauwe, 2011*; *Sage et al., 2012*, *2013*; *Schulze et al., 2013*) indicates that establishing a photorespiratory $CO_2$ pump was an early and important step in the evolution towards $C_4$ photosynthesis (*Figure 1C*). Since the two-carbon compound glycine serves as a transport metabolite, this photorespiratory $CO_2$ concentrating mechanism is also termed $C_2$ photosynthesis. Computational modeling of the evolutionary trajectory from $C_3$ to $C_4$ photosynthesis indicated $C_2$ photosynthesis represented an evolutionary intermediate state (*Heckmann et al., 2013*; *Williams et al., 2013*) as well suggesting that $C_2$ photosynthesis is a prerequisite for the evolution of $C_4$. However, it remained unclear if the evolution of $C_2$ photosynthesis fosters the evolution of $C_4$ photosynthesis beyond providing a selection pressure to reallocate Rubisco to the bundle sheath.

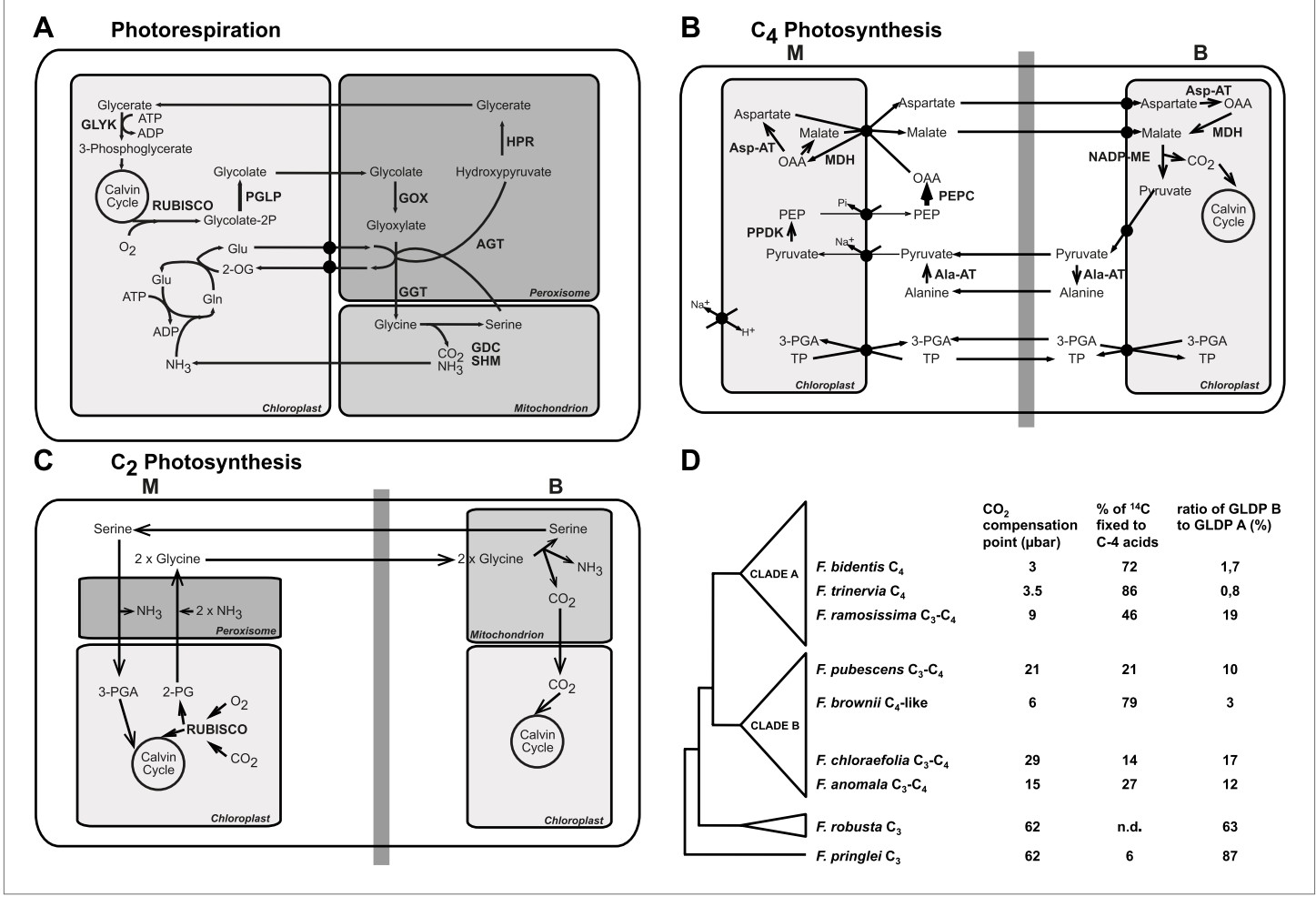

**Figure 1**. The genus *Flaveria* as a model organism to study $C_4$ evolution. Schematic view of the photorespiratory pathway (**A**), the NADP-ME type $C_4$ pathway as it can be found in $C_4$ Flaveria species (**B**) and the $C_2$ photosynthesis pathway (**C**). (**D**) Phylogeny and physiological properties of selected Flaveria species. The phylogeny was redrawn according to **McKown et al. (2005)**, $CO_2$ compensation points are taken from **Ku et al. (1991)**, incorporation of $^{14}CO_2$ is from **Moore et al. (1987)** and the ratios of GLDP B (expressed in all chlorenchyma cells) and GLDP A (expressed in bundle sheath cells only) are from **Schulze et al. (2013)**. (Abbreviations: AGT: serine glyoxylate aminotransferase; AlaAT: alanine aminotransferase; AspAT: aspartate aminotransferase; GDC: glycine decarboxylase complex; GGT: glutamate, glyoxylate-aminotransferase; GLYK: D-glycerate 3-kinase; GOX: glycolate oxidase; HPR: hydroxypyruvate reductase; MDH: malate dehydrogenase; NADP-ME: NADP dependent malic enzyme; PEPC: phosphoenolpyruvate carboxylase; PGLP: 2-phosphoglycerate phosphatase; PPDK pyruvate, phosphate-dikinase; RUBISCO: Ribulose-1,5-bisphosphat-carboxylase/-oxygenase; SHM: serine hydroxymethyltransferase; 2-OG: oxoglutarate; 2-PG 2-phosphoglycolate; 3-PGA: 3-phosphoglycerate; Gln: glutamine; Glu: glutamate; OAA: oxaloacetate; PEP: phosphoenolpyruvate; TP: triosephosphate).

In the present study, we have used the genus *Flaveria* as a model system for investigating the transition from $C_2$ to $C_4$ photosynthesis. To this end, we study a phylogenetic framework consisting of $C_3$, $C_3$–$C_4$ intermediate, and $C_4$ species (*Powell, 1978*; *Edwards and Ku, 1987*; *Ku et al., 1991*) of this genus which rather recently evolved $C_4$ (*Christin et al., 2011*), focusing on genes encoding photorespiratory enzymes and other components of $C_2$ photosynthesis. The genus *Flaveria* contains three main phylogenetic groups, of which the first diverging group includes all $C_3$ *Flaveria*. Clade B contains seven $C_3$–$C_4$ intermediate species and the $C_4$-like species *F. brownii*. All $C_4$ *Flaveria* species belong to clade A, which also contains several $C_4$-like species and the $C_3$–$C_4$ intermediate *F. ramosissima* (*McKown et al., 2005*; *Figure 1D*). We hypothesized that the analysis of species in the genus *Flaveria* combined with *in silico* modeling elucidates the evolutionary changes accompanying and following the establishment of the $C_2$ pathway. To this end we simulated the metabolism of $C_2$ plants by coupling a mechanistic model of $C_3$–$C_4$ intermediate photosynthesis (*von Caemmerer, 2000*; *Heckmann et al., 2013*)

with a detailed modified stoichiometric model of $C_4$ photosynthesis (*Dal'Molin et al., 2010*), and investigated the evolution of $C_4$ photosynthesis and photorespiration by following the changes in mRNA and protein abundance along the evolutionary path.

RNA and protein amounts of the majority of the photorespiratory enzymes were reduced in $C_4$ as compared to $C_3$ species. In contrast, photorespiratory mRNA and protein amounts did not decrease in the $C_3$–$C_4$ intermediate species but were mostly equal or even higher than in the $C_3$ species, demonstrating that the establishment of the photorespiratory $CO_2$ pump in the genus *Flaveria* relies on coordinated changes in the expression of all core photorespiratory enzymes. Metabolic modeling in combination with comparisons of transcript abundances in the different *Flaveria* species strongly indicates that introduction of $C_2$ photosynthesis has a direct impact on the nitrogen metabolism of the leaf. Its implementation necessitates the parallel establishment of components of the $C_4$ cycle to cope with these changes in refixation of photorespiratory nitrogen. Based on these results, we predict a mechanistic interaction between $C_4$ and $C_2$ photosynthesis.

## Results

### Selection of *Flaveria* species, cultivation of plant material and experimental design

To study the evolution of the expression of photorespiratory and $C_4$ cycle genes during the transition from $C_3$ to $C_4$ photosynthesis in the genus *Flaveria,* nine species reflecting the evolutionary trajectory taken were selected, including two $C_3$ (*F. robusta* and *F. pringlei*), two $C_4$ (*F. bidentis* and *F. trinervia*), and five $C_3$–$C_4$ intermediate species (*Figure 1D*). According to their $CO_2$ compensation points and the percentage of carbon initially fixed into malate and aspartate, *F. chloraefolia* and *F. pubescens* were earlier classified as type I $C_3$–$C_4$ intermediates. *F. anomala* and *F. ramosissima* belong to the type II $C_3$–$C_4$ intermediates and *F. brownii* is classified as a $C_4$-like species (*Edwards and Ku, 1987*; *Moore et al., 1987*; *Cheng et al., 1988*; *Ku et al., 1991*). Type I $C_3$–$C_4$ intermediates are defined as solely relying on the photorespiratory $CO_2$ concentration cycle whereas a basal $C_4$ cycle activity is present in type II $C_3$–$C_4$ intermediates species. $C_4$-like species exhibit much higher $C_4$ cycle activities but lack complete bundle sheath compartmentation of Rubisco activity (*Edwards and Ku, 1987*).

Four independent experiments with plants grown during different seasons were performed to identify differences between the species that are dependent on their different modes of photosynthesis and independent of environmental influences. For each experiment the plants were seeded concurrently and grown side-by-side under greenhouse conditions. The second and fourth visible leaves from the top of all nine species were harvested at noon on the same day for transcript and protein analysis. Plants for experiment one were harvested in September 2009, for experiment two in June 2010, for experiment three in October 2010 and for experiment four in April 2011. The amounts of the core photorespiratory and $C_4$ enzymes were assessed by immunoblotting using specific antibodies raised against synthetic peptides or recombinant proteins. The abundances of the corresponding RNAs as well of $C_4$ cycle associated transcripts were quantified by total transcriptome sequencing.

### The transcript profiles of the individual *Flaveria* species were comparable throughout all four experiments

The transcriptomes of the different *Flaveria* species were sequenced by Illumina technology following standard procedures. In total, close to 200 Gb of raw sequence data were produced. After filtering of low quality reads 30 to 58 million reads per species and experiment were quantified (*Figure 2—source data 1*). In a cross species approach, we mapped the sequences onto the minimal set of *Arabidopsis thaliana* coding sequences using the BLAST-like alignment tool BLAT (*Kent, 2002*) as described previously (*Gowik et al., 2011*) (*Figure 2—source data 2*, data available from the Dryad Digital Repository: 10.5061/dryad.q827h). We were able to align approx. 50% of our reads to the *Arabidopsis* transcripts. This is lower as compared to a similar approach using 454 sequencing (*Gowik et al., 2011*) and likely due to the shorter read length of the Illumina compared to the 454 reads. To overcome the low mapping efficiency, the leaf transcriptomes of *Flaveria* species were assembled de novo based on 454 (*Gowik et al., 2011*) and Illumina reads (this study). Among the contigs from *F. robusta*, we identified full-length transcripts for all photorespiratory and $C_4$ genes in the focus of the present study and used these for further read mapping and detailed analysis.

To evaluate the variation between the four independent experiments, we performed hierarchical sample clustering and a principal component analysis of the transcript profiles derived from read mapping on the minimal set of *Arabidopsis* coding sequences. Hierarchical sample clustering using Pearson correlation and average linkage clustering shows that the transcript profiles of all *Flaveria* species were quite similar in all four experiments since the samples cluster strictly species-wise (*Figure 2A*). The transcriptome patterns are influenced by the photosynthesis type and the phylogenetic relationships of the different species. The two $C_4$ species, both belonging to clade A, cluster together as do the two $C_3$ species that belong to the basal *Flaveria* species. Within the $C_3$–$C_4$ intermediates the two more advanced intermediates *F. ramosissima* and *F. anomala* cluster together, the only pattern which contradicts phylogenetic proximity since *F. ramosissima* belongs to clade A and *F. anomala* belongs to clade B. The last cluster consists out of the $C_3$–$C_4$ intermediates *F. chloraefolia* and *F. pubescens*, and the $C_4$-like species *F. brownii*.

Principle component analysis supports the results of the hierarchical clustering. The samples are mainly separated by photosynthesis type and phylogenetic relationships with the two intermediate species from different phylogenetic trajectories again forming a tight cluster (*Figure 2B*). The first three components, shown in *Figure 2B*, explain only 27% of the total variance. This is in good accordance with earlier results where it was shown that about 16% of all analyzed genes showed photosynthesis type related expression changes when the transcriptomes of the $C_4$ species *F. trinervia* and *F. bidentis*, the $C_3$ species *F. robusta* and *F. pringlei* and the $C_3$–$C_4$ intermediate species *F. ramosissima* were compared (*Gowik et al., 2011*).

## Amounts of photorespiratory transcripts and proteins indicate that the $C_2$ pathway was established early during $C_4$ evolution in *Flaveria* and is present also in the $C_4$-like species *F. brownii*

Photorespiratory genes are expressed in all species and photorespiratory proteins are detected in all species. To visualize the differences in transcript and protein abundance heat maps were plotted (*Figure 3*). The transcription of all photorespiratory genes except the transport proteins DIT1 and DIT2 and one isoform of GLDH was downregulated in the $C_4$ species *F. bidentis* and *F. trinervia* compared to the $C_3$ species *F. pringlei* und *F. robusta* (*Figure 3A*, *Figure 3—source data 1*). Both dicarboxylate transporters play an important role in generating the transfer acids in the $C_4$ pathway of NADP-ME plants such as *F. trinervia* and *F. bidentis* (*Renne et al., 2003*; *Gowik et al., 2011*; *Kinoshita et al., 2011*). This may explain why their expression pattern is more similar to the $C_4$ genes than to the other photorespiratory genes.

The amounts of photorespiratory transcripts did not decrease gradually from $C_3$ to $C_4$ but the expression levels in the $C_3$–$C_4$ intermediate species *F. chloraefolia*, *F. pubescens*, *F. anomala* and *F. ramosissima* were mostly equal or higher than in the $C_3$ species. An exception are the transcripts of one GLDP, one GLDH and one SHM isoform which are drastically down-regulated also in the $C_3$–$C_4$ intermediate species. It was shown earlier that the down-regulation of this GLDP isoform is tightly associated with the establishment of the $C_2$ pathway in *Flaveria* (*Schulze et al., 2013*). The down-regulation of the GLDH and SHM isogenes might have similar reasons since both enzymes are also involved in glycine decarboxylation. Only the $C_4$-like species *F. brownii* is intermediate with respect to photorespiratory transcripts. 19 of 27 transcripts are reduced compared to the $C_3$–$C_4$ intermediate and $C_3$ species but have higher levels than the true $C_4$ species *F. bidentis* and *F. trinervia*. Exceptions are the components of the glycine decarboxylase complex as the respective transcripts levels are equal to these in the $C_3$ and $C_3$–$C_4$ intermediate species (*Figure 3A*).

The expression patterns described above were not only found for the genes encoding the core enzymes of photorespiration but also for the genes responsible for recycling of ammonia set free during photorespiration, GS/GOGAT. Also the genes of recently discovered transporters associated with photorespiration, PLGG1 and BOU (*Eisenhut et al., 2013*; *Pick et al., 2013*), behave accordingly.

To test whether transcript abundance reflects protein abundance, amounts of core photorespiratory proteins in the leaves of all nine species were quantified by protein gel blots. To this end we generated antibodies against conserved peptides from *Flaveria* GLDP, GLDT, GLDL, SHM, HPR, PGLP and GLYK proteins. Total leaf proteins were extracted from plant material harvested together with the material used for RNA isolation and equal amounts of protein were separated via SDS gelelectrophoresis prior to blotting (*Figure 3—figure supplement 1*). The changes of protein amounts essentially reflected the changes of the amounts of the corresponding transcripts (*Figure 3B*,

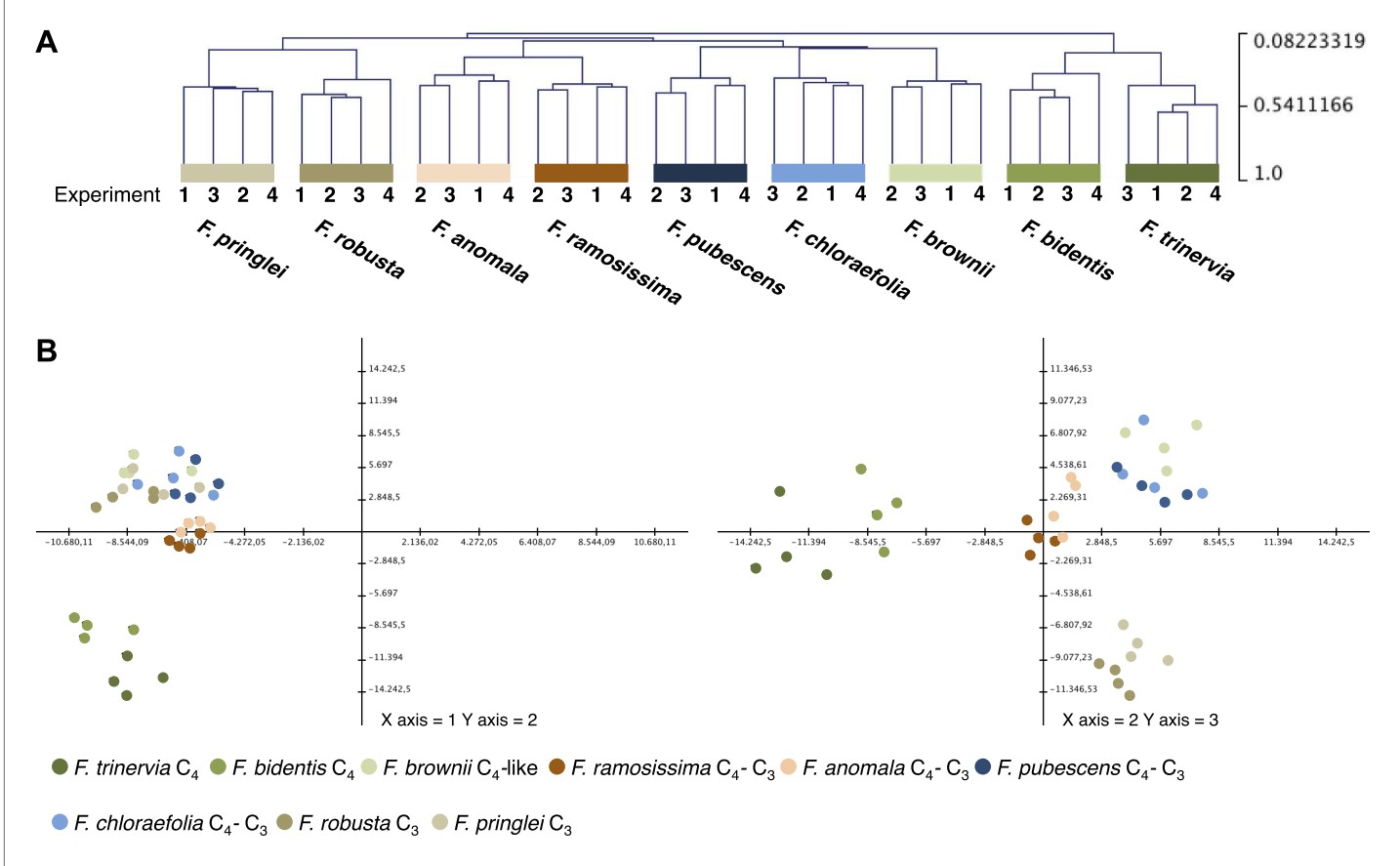

**Figure 2**. Variation of transcript profiles of the individual *Flaveria* species between the four experiments. (**A**) Hierarchical sample clustering of all expressed transcripts. The tree was calculated with the MEV program using the HCL module with Pearson correlation and the average linkage method. (**B**) Principal component analysis of transcript levels. The first three components explain 27% of the total variance.

The following source data are available for figure 2:

**Source data 1**. Results of the Illumina sequencing and cross species read mapping.

**Source data 2**. Quantitative information for all reads mapped in a cross species approach onto the reference transcriptome from *Arabidopsis thaliana*.

*Figure 3—figure supplement 2*, *Figure 3—source data 2*). The amounts of core photorespiratory proteins in the $C_3$–$C_4$ intermediates were equal to the amounts in the $C_3$ species. A clear reduction of these proteins can be observed only for the true $C_4$ species and the $C_4$–like species. *F. brownii* exhibits intermediate amounts of most photorespiratory proteins. This indicates that the regulation of photorespiratory genes mainly occurs on the transcriptional level and that our approach to analyze the photorespiratory activity by comparative transcriptomics is reasonable.

While the overall patterns remain similar between all independent experiments, individual proteins and transcripts vary between the four experiments. This likely reflects adjustments of photorespiratory gene expression to the different light and temperature conditions in our green house in the different seasons of the year.

We conclude that the four experiments support the establishment of a photorespiratory $C_2$ cycle early during $C_4$ evolution in *Flaveria* and that this $C_2$ cycle was maintained until Rubisco activity was constricted to the bundle sheath cells in the true $C_4$ *Flaveria* species.

### An integrated model of $C_2$ photosynthesis

While the principal physiological differences between $C_3$ and $C_4$ leaves are widely understood, knowledge about the metabolic reconfiguration required to implement a functional $C_2$ pathway into a $C_3$ leaf is

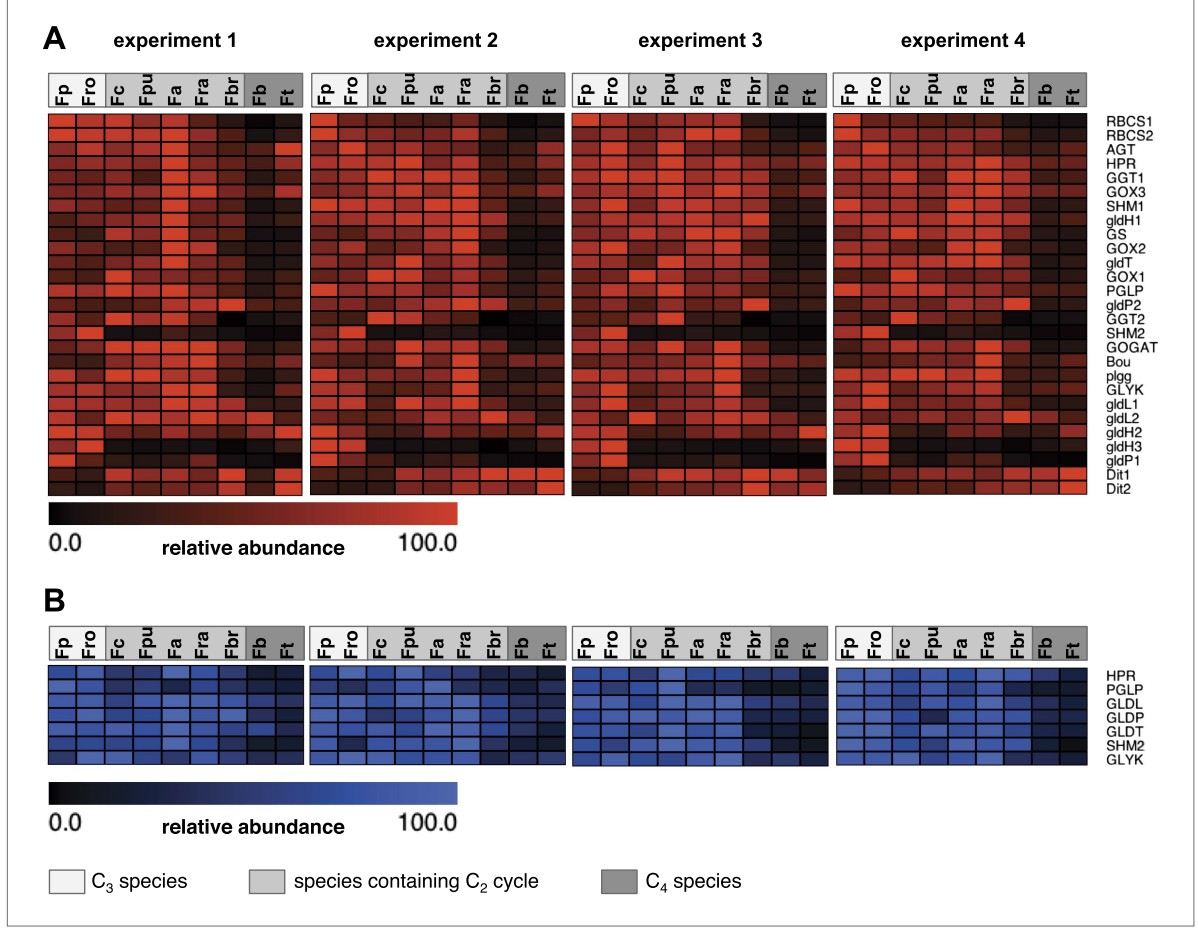

**Figure 3**. Abundance of photorespiratory transcripts and proteins in leaves of individual *Flaveria* species. Normalized transcript (**A**) and protein (**B**) amounts are plotted as heat maps. Transcript amounts were determined by Illumina sequencing of the leaf transcriptomes and read mapping on selected *F. robusta* full length transcript sequences. Protein amounts were determined by protein gel blots. See *Figure 3—source data 1* for absolute transcript levels, *Figure 3—source data 2* for protein quantification and *Figure 3—figure supplements 1 and 2* for immunoblots. Fp: *F. pringlei* ($C_3$); Fro: *F. robusta* ($C_3$); Fc: *F. chloraefolia* ($C_3$–$C_4$); Fpu: *F. pubescens* ($C_3$–$C_4$); Fa: *F. anomala* ($C_3$–$C_4$); Fra: *F. ramosissima* ($C_3$–$C_4$); Fbr: *F. brownii* ($C_4$-like); Fb: *F. bidentis* ($C_4$); Ft: *F. trinervia* ($C_4$).

The following source data and figure supplements are available for figure 3:

**Source data 1**. Transcript abundance of photorespiratory genes determined by read mapping on *F. robusta* full length transcript sequences.

**Source data 2**. Quantification of photorespiratory proteins by protein gel blot.

**Figure supplement 1**. Results of the protein analyses.

**Figure supplement 2**. Results of the protein analyses.

incomplete. In particular, moving glycine from mesophyll to bundle sheath cells (*Hylton et al., 1988*; *Morgan et al., 1993*) does not only translocate carbon, it also transports one nitrogen atom per two carbon atoms. Evidently, implementing the $C_2$ carbon pump requires balancing of metabolic routes to maintain homeostasis of both carbon and nitrogen metabolism (*Monson and Rawsthorne, 2000*). How this can be achieved is non-intuitive and it thus requires a systematic analysis by metabolic modeling. To this end, we simulated the leaf metabolism of a $C_2$ plant using an integrated model. We coupled a mechanistic model of $C_3$–$C_4$ intermediate photosynthesis (*von Caemmerer, 2000*; *Heckmann et al., 2013*) with a modified genome-scale stoichiometric model of $C_4$ photosynthesis that was designed to describe the entire metabolic interactions of mesophyll and bundle sheath cells in $C_4$ leaves (*Dal'Molin et al., 2010*).

We used the mechanistic model to predict constraints for the stoichiometric model. It provided values for net $CO_2$ uptake, Rubisco carboxylation as well as oxygenation in mesophyll and bundle sheath, $CO_2$ leakage from the bundle sheath, PEPC activity in the mesophyll, activity of NADP-ME in the bundle sheath, plasmodesmatal flux of glycine and serine, and decarboxylation by the GDC. Given specific activities of the $C_2$ and $C_4$ cycles in the mechanistic model, we used flux balance analysis (FBA) to predict detailed flux distributions that follow biologically realistic optimality criteria (*Varma and Palsson, 1994*). We employed a maximization of leaf biomass production, followed by a minimization of the sum of absolute fluxes including transport processes. In the minimization of total flux, we allocated higher weights to plasmodesmatal fluxes in order to account for the trade-off between $CO_2$ leakage and diffusion of metabolites between the cells. This framework allows us to investigate the most parsimonious implementation of $C_2$ and $C_4$ cycles, given a hypothesis about which metabolites are suitable for plasmodesmatal transport.

The first outcome of simulating the photorespiratory $CO_2$ pump was that the establishment of the $C_2$ pathway has indeed a direct impact on the nitrogen metabolism of the leaf. It transports two molecules of glycine from the mesophyll to the bundle sheath, where one molecule each of serine, $CO_2$, and ammonium are produced. $CO_2$ is fixed by bundle sheath Rubisco and serine is transferred back to the mesophyll, where it is used for the regeneration of phosphoglycerate and photorespiratory glycine. This results in a net transport of $CO_2$ but also ammonia from the mesophyll to the bundle sheath. To create a noticeable $CO_2$ enrichment in the bundle sheath, the $C_2$ cycle must run with an appreciable capacity; indeed, the mechanistic model of $C_3$–$C_4$ intermediate photosynthesis predicted an oxygenation rate of Rubisco of about one third of its carboxylation rate. Running at such rates, the $C_2$ cycle will create a massive nitrogen imbalance between mesophyll and bundle sheath cells, as was also predicted earlier by *Monson and Rawsthorne (2000)*. Within the stoichiometric model, the free diffusion of ammonia between the two cell types was not allowed, since ammonia is toxic and known to effectively uncouple electrochemical gradients (*Krogmann et al., 1959*). Thus, ammonia must be refixed in the bundle sheath cells and transferred back to the mesophyll in the form of amino acids. According to the intergrated model, ammonia is fixed by glutamine synthetase and glutamine oxoglutarate aminotransferase (GS/GOGAT) in the bundle sheath cells (*Figure 4*). Consistent with this prediction, we found that GS/GOGAT transcripts were upregulated in the $C_3$–$C_4$ intermediate species (*Figure 3*).

Estimating whether a certain metabolite is suitable for maintaining a diffusional gradient between mesophyll and bundle sheath is an unsolved problem. The impact on regulatory mechanisms and homeostasis of the $C_3$ leaf may render some metabolites unsuitable to serve as transport metabolites. We address this problem by modeling multiple scenarios that assume different transport metabolites.

If major amino acids and the corresponding oxoacids and dicarbonic acids are allowed to freely diffuse between cells in an integrated model representing a $C_2$ cycle, glutamate is predicted to be transferred to the mesophyll, where it is deaminated by GGT, regenerating the photorespiratory glycine. The resulting 2-oxoglutarate is transferred back to the bundle sheath cells (*Figure 4A*). The model preference for glutamate/2-oxoglutarate reflects the minimization of total flux in the FBA model, as this effectively minimizes the number of active enzymatic reactions and holds the plasmodesmatal flux for ammonia balance at one acceptor and one transport metabolite.

To elucidate if alternative solutions exist that contain more steps but retain the same biomass output, the 2-oxoglutarate transfer between mesophyll and bundle sheath was constrained to prevent the glutamate/2-oxoglutarate exchange. The integrated model then predicts an alanine/pyruvate shuttle (*Figure 4B*). The glutamate produced by GS/GOGAT activity in the bundle sheath cells is used by alanine aminotransferase (Ala-AT) to aminate pyruvate. The resulting alanine is transferred to the mesophyll and trans-aminated by Ala-AT resulting in pyruvate and glutamate. The glutamate is used to regenerate photorespiratory glycine and pyruvate is transferred back to the bundle sheath.

If alanine and pyruvate transfer are also constrained, the model predicts an aspartate/malate shuttle (*Figure 4C*). This includes the oxidation of malate in the bundle sheath. The resulting oxaloacetate (OAA) is aminated by aspartate aminotransferase (Asp-AT) and aspartate moves to the mesophyll. Here aspartate is trans-aminated by Asp-AT and malate is regenerated by reduction of the resulting OAA and transferred to the bundle sheath.

In all these scenarios, further increasing the weights on plasmodesmatal flux leads to transporter metabolites with increased N carrying capacity such as asparagine (*Figure 4—source data 1*).

In a restrictive scenario, all nitrogen containing compounds were excluded from plasmodesmatal transport, except for glycine and serine, which are used by the $C_2$ cycle itself. In this case, the model

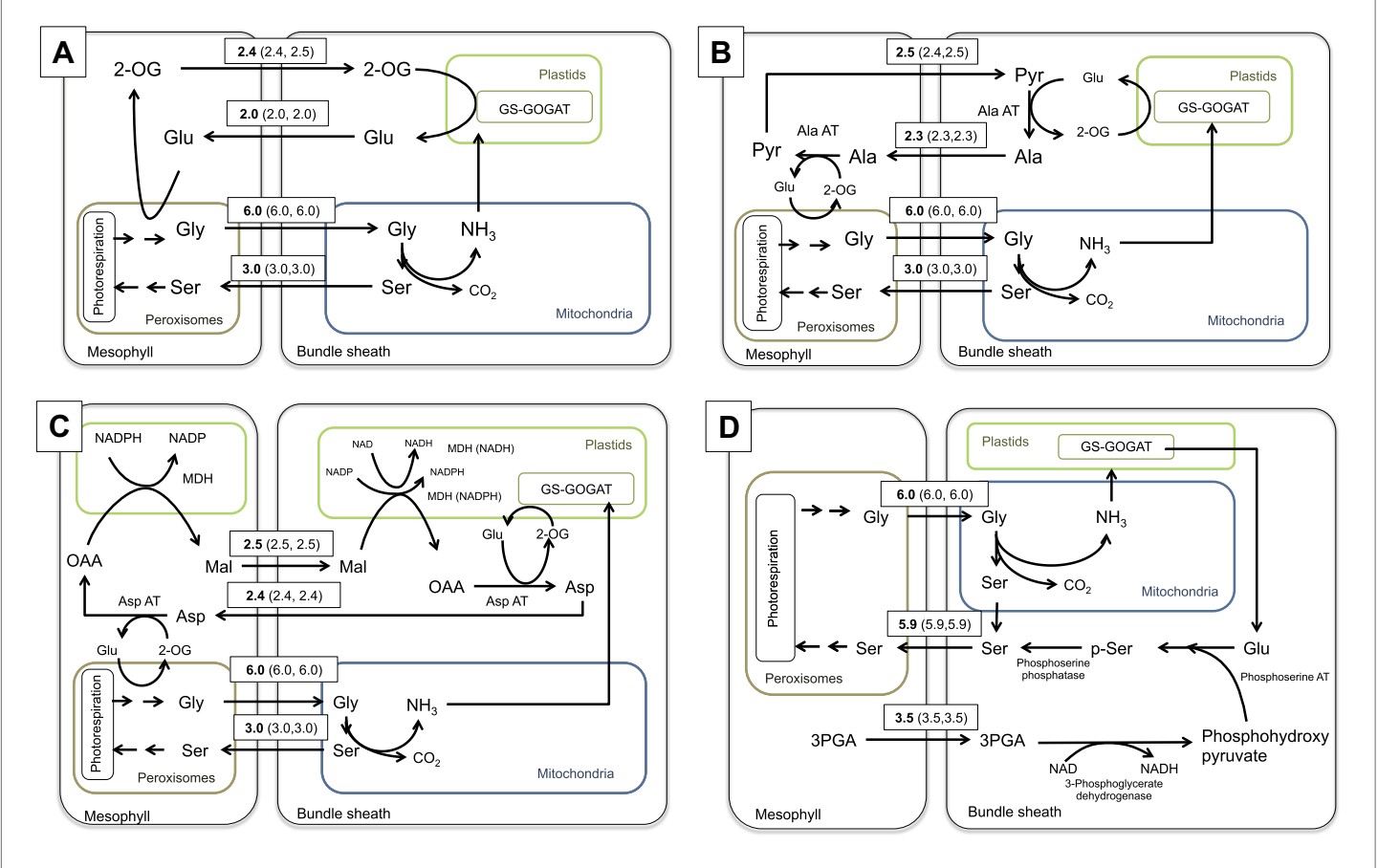

**Figure 4**. Flux Balance Analysis of the $C_2$ photosynthetic pathway. Predicted fluxes if (**A**) major amino acids and the corresponding oxoacids and dicarbonic acids are allowed to freely diffuse between cells, (**B**) the α-ketoglutarate and glutamate transfer between mesophyll and bundle sheath was constrained (**C**) additionally the transfer of alanine and pyruvate between mesophyll and bundle sheath was constrained (**D**) transfer of all nitrogen containing compounds except for glycine and serine, which are used by the $C_2$ cycle were constrained. Fluxes are given in µmol $s^{-1}$ $m^{-2}$. Values in brackets show minimum and maximum of flux resulting from flux variability analysis. Flux of dissolved gasses, sucrose, inorganic compounds and processes that carry flux below 1 µmol $s^{-1}$ $m^{-2}$ are not shown. The sums of absolute fluxes over the plasmodesmata for the different variants were (**A**): 17.8 µmol $s^{-1}$ $m^{-2}$; (**B**): 18.4 µmol $s^{-1}$ $m^{-2}$; (**C**): 19.0 µmol $s^{-1}$ $m^{-2}$; (**D**): 22.1 µmol $s^{-1}$ $m^{-2}$. See *Figure 4—source data 1* for plasmodesmatal fluxes.

The following source data are available for figure 4:

**Source data 1**. Fluxes over plasmodesmata depending on the weight on plasmodesmatal fluxes including flux variability analysis.

predicts that bundle sheath derived ammonia is transferred from glutamate to phosphohydroxy-pyruvate by phosphoserine aminotransferase to yield phosphoserine; phosphoserine is then converted to serine by phosphoserine phosphatase. Finally, the serine moves to the mesophyll. This variant includes the transfer of 3-phosphoglycerate from the mesophyll to the bundle sheath, where it is converted to phosphohydroxy pyruvate by 3-phosphoglycerate dehydrogenase (*Figure 4D*).

## The model predicts a mechanistic interaction between $C_2$ and $C_4$ cycle

In $C_3$ plants, basal activities of the typical $C_4$ cycle enzymes are present (*Aubry et al., 2011*). When our integrated model is parameterized to include an active $C_4$ cycle, it predicts that a contingent of the bundle sheath ammonia will be transferred to the mesophyll cells by the $C_4$ cycle as a biomass neutral alternative to the 2-OG/Glu shuttle or as the unique solution when additional weight on plasmodesmatal fluxes is applied (*Figure 5—source data 1*). In this solution malate is decarboxylated in the bundle sheath cells. $CO_2$ is refixed by Rubisco, and the resulting pyruvate is aminated by Ala-AT. Alanine moves to the mesophyll cells, where ammonia is fed into the photorespiratory cycle by Ala-AT

and GGT. The resulting pyruvate is converted back to malate by PPDK, PEPC, and NADPH-dependent MDH (*Figure 5A*). Flux variability analysis shows that only marginal variability in the fluxes of the shuttle is possible (*Figure 5—source data 1*). According to our model predictions, the cycle is active even at low PEPC activities, such as those measured in $C_3$ *Flaveria* species (*Gowik et al., 2011*; *Heckmann et al., 2013*). When the $C_4$ cycle runs with low capacity, according the model, the surplus of bundle sheath ammonia is transferred back to the mesophyll by the glutamate/2-oxoglutarate shuttle. Once the capacity of the $C_4$ cycle gradually increases, the recirculation of nitrogen is shifted from the glutamate/2-oxoglutarate shuttle towards the $C_4$ cycle (*Figure 5B*). The predicted biomass production increases linearly with $C_4$ cycle activity (*Figure 5C*). Thus, our model predicts a strong interaction between $C_2$ and $C_4$ photosynthesis.

## Analysis of $C_4$ cycle gene expression in $C_3$–$C_4$ intermediate *Flaveria* species

When the $C_2$ cycle is running with high capacity, our integrated modeling approach predicts the necessity of auxiliary metabolite fluxes between mesophyll and bundle sheath cells to prevent a massive nitrogen imbalance. Among those auxiliary fluxes were the pyruvate/alanine and the malate/aspartate exchanges. The metabolites used in these shuttles also serve as transport metabolites in $C_4$ photosynthesis. Furthermore, the model highlights the possibility that a low capacity $C_4$ cycle balances part of the $C_2$ cycle ammonia production. Therefore we analyzed in detail the expression of $C_4$ cycle related genes in our dataset. True $C_4$ *Flaverias,* such as *F. bidentis* or *F. trinervia*, are believed to use a NADP-ME type $C_4$ cycle (*Moore et al., 1984*; *Ku et al., 1991*; *Meister et al., 1996*; *Gowik et al., 2011*). All genes associated with this type of $C_4$ photosynthesis are gradually upregulated in the analyzed $C_3$–$C_4$ intermediate species in line with their degree of 'C$_4$-ness'. This is true for the typical $C_4$ enzymes like PEPC, PPDK, MDH, NADP-ME, Ala-AT and a plastidic aspartate aminotransferase (Asp-AT), as well as for several $C_4$ associated transporters, such as the pyruvate transporter BASS2, the H$^+$/Na$^+$ exchanger NHD, the PEP translocator CUE1 and the putative malate and aspartate transporters DIT1 and DIT2 (*Weber and von Caemmerer, 2010*; *Brautigam et al., 2011*; *Furumoto et al., 2011*; *Gowik et al., 2011*). The regulators of the $C_4$ enzymes (like PEPC kinase or the PPDK regulatory protein) and enzymes with auxiliary functions of $C_4$ enzymes (like pyrophosphatases or adenosinmonophosphatases) show a similar pattern (*Figure 6*, *Figure 6—source data 1*). To corroborate the results of the transcript abundance measurements, selected $C_4$ cycle enzymes (PEPC, PPDK, and NADP-ME) were measured by immunoblotting. The protein abundance correlates well with the transcript abundance (*Figure 6*, *Figure 6—source data 2*).

The expression changes of $C_4$ cycle genes do not all follow the same quantitative pattern (*Figure 6C*). Although all of these genes gradually increase in expression when plants gain $C_4$ properties, as judged, for example, by the percentage of $^{14}CO_2$ directly fixed into $C_4$ acids (*Vogan and Sage, 2011*), the quantitative changes in gene expression are quite different. PEPC and PPDK transcript amounts increase slowly in the $C_3$–$C_4$ intermediates *F. chloraefolia*, *F. pubescens*, and *F. anomala*, more steeply in the advanced $C_3$–$C_4$ intermediate *F. ramosissima* and the $C_4$-like species *F. brownii* before reaching the highest transcript abundances in the true $C_4$ species (*Figure 6C*). In contrast, NADP-ME and Ala-AT gene expression already increase in expression in the more $C_3$-like intermediate species. Their expression rises more linearly in the further advanced intermediates and plateaus in the $C_4$-like and $C_4$ species. If one uses the different *Flaveria* species as evolutionary proxies as suggested by the results of *Heckmann et al. (2013)*, these results suggest that NADP-ME and Ala-AT are strongly upregulated earlier in evolution than other $C_4$ core enzymes like PEPC or PPDK.

*F. chloraefolia* is classified as a type I $C_3$–$C_4$ intermediate species, and no enhanced $C_4$ cycle activity should be present in this species based on the classification. We detected upregulation of all NADP-ME type associated $C_4$ genes, with some of the genes showing comparable small increases in expression (*Figure 6*). This is in line with the results of $^{14}CO_2$ uptake studies that indicate about 14% of $CO_2$ is directly incorporated into $C_4$ acids in *F. chloraefolia*, whereas only 6% goes into $C_4$ acids directly in the $C_3$ species *F. pringlei* (*Moore et al., 1987*). We think therefore that a basal $C_4$ cycle activity is present in *F. chloraefolia* and its classification as type I $C_3$–$C_4$ intermediate is questionable.

A gene encoding a mitochondrial NAD dependent malate dehydrogenase as well as several cytosolic and especially one mitochondrial Asp-AT were upregulated exclusively in the $C_3$–$C_4$ intermediate species and the $C_4$-like *F. brownii* (*Figure 6*). Often, high activities of these genes are associated with the NAD-ME or PEP-CK type of $C_4$ photosynthesis. NAD dependent malic enzyme and PEP

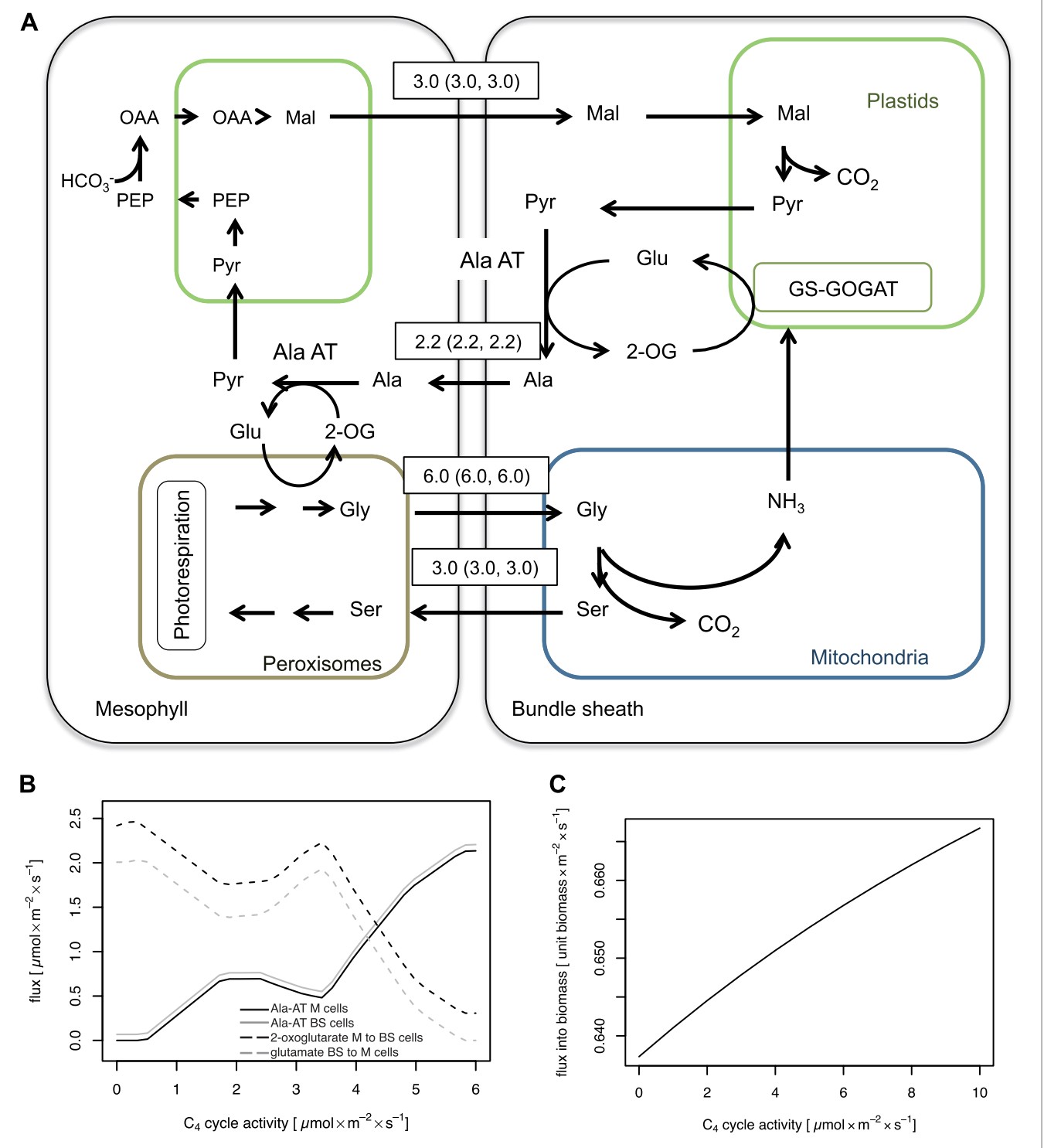

**Figure 5**. Mechanistic interaction between $C_2$ and $C_4$ cycle. (**A**) Predicted fluxes when the model is parameterized to include activity of the $C_4$ cycle enzymes. Fluxes are given in μmol s$^{-1}$ m$^{-2}$. Values in brackets show minimum and maximum of flux resulting from flux variability analysis. The sum of absolute flux over plasmodesmata was 21.9 μmol s$^{-1}$ m$^{-2}$. Flux of dissolved gasses, sucrose, inorganic compounds and processes that carry flux below 1 μmol s$^{-1}$ m$^{-2}$ are not shown. See *Figure 5—source data 1* for plasmodesmatal fluxes. (**B**) Predicted activities of Ala-AT in mesophyll (black line) and bundle sheath (gray line) cells and predicted transfer of α-ketoglutarate from mesophyll to bundle sheath cells (black dashed line) and glutamate from
*Figure 5. Continued on next page*

*Figure 5. Continued*

bundle sheath to mesophyll cells (gray dashed line) at low $C_4$ cycle activities. (**C**) Changes in biomass production with varying (low) activity of the $C_4$ cycle in a $C_2$ plant.

The following source data are available for figure 5:

**Source data 1**. Fluxes over plasmodesmata depending on the weight on plasmodesmatal fluxes including flux variability analysis.

carboxykinase genes were only very lowly expressed in all analyzed *Flaverias*, and no obvious differences between the $C_3$–$C_4$ intermediates and the other species could be found (***Figure 2—source data 2***, data available from the Dryad Digital Repository: http://dx.doi.org/10.5061/dryad.q827h).

We found no transcriptomic evidence that ammonia is recirculated by the phosphoserine pathway predicted by the model that restricts the free diffusion of all amino acids except serine and glycine. The amounts of transcripts for all three enzymes of this pathway, i.e., phosphoserine aminotransferase, phosphoserine phosphatase, and 3-phosphoglycerate dehydrogenase, were found to be very low in all analyzed *Flaveria* species (***Figure 2—source data 2***, data available from the Dryad Digital Repository: http://dx.doi.org/10.5061/dryad.q827h) (***Mallmann et al., 2014***).

Taken together, these data imply that the anaplerotic ammonia shuttle, required to maintain the nitrogen homeostasis in mesophyll and bundles sheath cells of plants performing $C_2$ photosynthesis, is active in all analyzed $C_3$–$C_4$ *Flaveria* species, as predicted by the computer simulations. Furthermore, it appears that even the most $C_3$-like $C_3$–$C_4$ intermediate species analyzed within the present study, *F. chloraefolia*, exhibits low level $C_4$ cycle activity. This activity is again in accordance with the *in silico* model, which predicts the $C_4$ cycle to be a highly efficient ammonia recirculation pathway.

## Discussion

Photorespiration is mainly seen as a wasteful process, which arises from a malfunction of Rubisco and reduces photosynthetic efficiency (***Ogren, 1984***). In a high $CO_2$ atmosphere, Rubisco can operate efficiently. But the current atmospheric $CO_2$ concentration, combined with heat and drought, leads to an enhanced oxygenase activity and thereby the photosynthetic efficiency decreases (***Raines, 2011***). Up to 30% of the initially fixed $CO_2$ may be lost by photorespiration (***Bauwe et al., 2010***). $C_4$ plants avoid this problem by enriching $CO_2$ at the site of Rubisco. $CO_2$ is prefixed in the mesophyll and released in the bundle sheath cells, where Rubisco is operating (***Hatch, 1987***). The establishment of the photorespiratory $CO_2$ pump, which relocates the release of photorespiratory $CO_2$ to the bundle sheath cells, appears to be an important intermediate step towards the $C_4$ cycle and our detailed study of *Flaveria* intermediate species suggests that genes associated with $C_4$ photosynthesis also played a role in the $C_2$ cycle.

### Implementation of the $C_2$ pathway leads to high expression of photorespiratory genes in $C_3$-$C_4$ intermediate *Flaveria* species

The expression of photorespiratory genes, including all genes encoding the core enzymes of the pathway, most of the transporters, and the enzymes involved in ammonia refixation, is not downregulated in the analyzed intermediate species; the transcript and protein amounts remain constant or in some cases are even higher compared to $C_3$ species. A significant drop in photorespiratory gene expression is only observed in the $C_4$-like species *F. brownii* and is decreased further in the $C_4$ species. Together with earlier results (***Schulze et al., 2013***), this indicates that indeed a $C_2$ photosynthetic cycle is active in all these $C_3$-$C_4$ intermediate *Flaveria* species and that a reduction in photorespiratory transcripts and proteins only occurs once the amounts of Rubisco have been reduced in the mesophyll as was described for the $C_4$-like species *F. brownii* (***Bauwe, 1984***; ***Holaday et al., 1988***). Rubisco reduction in the mesophyll is thus a late step of $C_4$ evolution, which in the *Flaveria* series appears to not occur gradually but rather abruptly towards the end of the evolutionary trajectory. It is followed by a strong increase of $C_4$ cycle activity, as can be deduced from the upregulation of PEPC and PPDK genes in the real $C_4$ species (***Figure 6C***), when the primary $CO_2$ fixation is completely taken over by PEPC. In the intermediate species $C_2$ and $C_4$ cycles operate in parallel leading to similar or higher photorespiratory gene expression compared with the $C_3$ species.

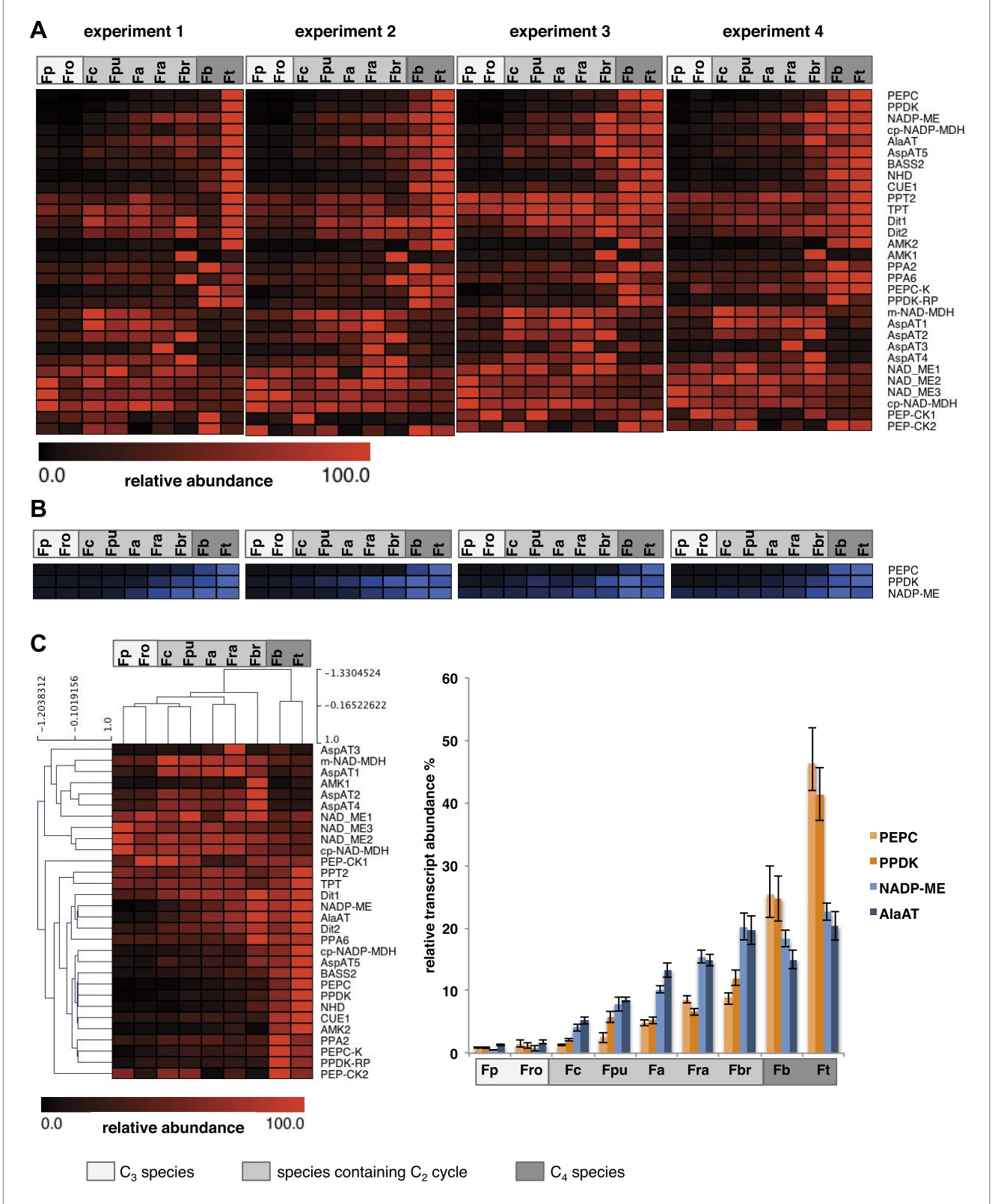

**Figure 6**. Abundance of C₄ related transcripts and proteins in leaves of individual Flaveria species. Normalized transcript (**A**) and protein (**B**) levels are plotted as heat maps. Transcript amounts were determined by Illumina sequencing of the leaf transcriptomes and read mapping on selected *F. robusta* full length transcript sequences. Protein amounts were determined by protein gel blots. See ***Figure 6—source data 2*** for absolute transcript level,
*Figure 6. Continued on next page*

*Figure 6. Continued*

*Figure 6—source data 2* for protein quantification and *Figure 3—figure supplement 1* for immunoblots. (**C**) Mean values of transcript levels from all four experiments were clustered by hierarchical using the HCL module of MEV program with Pearson correlation and the average linkage method. The relative transcript abundance for PEPC, PPDK, NADP-ME and Ala-AT (mean values from all four experiments) are plotted for all nine species. Fp: *F. pringlei* (C$_3$); Fro: *F. robusta* (C$_3$); Fc: *F. chloraefolia* (C$_3$–C$_4$); Fpu: *F. pubescens* (C$_3$–C$_4$); Fa: *F. anomala* (C$_3$–C$_4$); Fra: *F. ramosissima* (C$_3$–C$_4$); Fbr: *F. brownii* (C$_4$–like); Fb: *F. bidentis* (C$_4$); Ft: *F. trinervia* (C$_4$).

The following source data and figure supplements are available for figure 6:

**Source data 1**. Transcript abundance of C$_4$ cycle genes determined by read mapping on *F. robusta* full length transcript sequences.

**Source data 2**. Quantification of C$_4$ proteins by protein gel blots.

**Figure supplement 1**. Results of the protein analyses.

## Analysis of C$_4$ cycle gene expression supports the predictions of the C$_2$ model for C$_3$–C$_4$ intermediate *Flaveria* species and implies the early establishment of a complete C$_4$ pathway

The model of the C$_2$ cycle and the underlying metabolism proposes GS/GOGAT, Ala-AT, and Asp-AT to be involved in balancing the amino groups during C$_2$ cycle operation (*Figure 4*). The transcriptome data from the C$_3$-C$_4$ intermediate *Flaveria* species largely support the results of our integrated model for the C$_2$ pathway (*Figure 6*). In these species we found an upregulation of genes involved in the three most likely mechanisms for the recovery of ammonia predicted by the model. GS/GOGAT, which catalyzes the primary refixation of ammonia in the bundle sheath cells, is important for all three versions of ammonia shuttles (*Figure 4*) and is upregulated in the intermediate species. Transcripts for the glutamate/2-oxoglutarate shuttle, the alanine/pyruvate shuttle, and the aspartate/malate shuttle are enriched in all C$_3$–C$_4$ intermediates compared to the C$_3$ and C$_4$ *Flaverias*. For the alanine/pyruvate shuttle, Ala-AT is needed in the bundle sheath and the mesophyll cells. Ala-AT is upregulated already in the least advanced C$_3$–C$_4$ intermediates *F. chloraefolia* and *F. pubescens*, but also in all the other C$_3$–C$_4$ intermediates. Ala-AT transcripts are also highly abundant in the true C$_4$ *Flaverias* since Ala-AT is directly involved in the C$_4$ cycle when alanine is used as transport metabolite.

We found several Asp-AT and two MDH genes upregulated in the C$_3$–C$_4$ intermediate species (*Figure 6*). The chloroplast-located MDH and Asp-AT genes are involved in the C$_4$ cycle of C$_4$ *Flaverias*, in which malate and aspartate are used concurrently as C$_4$ transport metabolites (*Meister et al., 1996*). Two further Asp-AT genes and another MDH gene were found to be upregulated exclusively in the C$_3$–C$_4$ intermediates, including the C$_4$-like species *F. brownii*. The most likely reason for upregulation of these genes is their involvement in the recirculation of photorespiratory ammonia by a malate/aspartate shuttle.

The pathways of ammonia recirculation between mesophyll and bundle sheath foreshadow the establishment of a true C$_4$ cycle (*Figure 4*). All variants described above need the establishment of inter- and intra-cellular transport capacities for amino acids and small organic acids, which are also needed for a functional C$_4$ cycle (*Weber and von Caemmerer, 2010*). The existence of an aspartate/malate and an alanine/pyruvate shuttle anticipates important components of a functional C$_4$ pathway. Our transcript data imply that both of these shuttles are active in C$_3$–C$_4$ intermediate *Flaverias*. Only a few additions would be required to convert these pathways of ammonia recirculation into a C$_4$-like CO$_2$ concentration mechanism, that is, malate would have to be decarboxylated in the bundle sheath cells and pyruvate would have to be converted to malate in the mesophyll. Our transcript data implies that this conversion of the photorespiratory ammonia recirculation pathway into a C$_4$-like CO$_2$ concentrating pump must have been an early event in C$_4$ evolution of *Flaveria* since already in the least advanced intermediates such as *F. chloraefolia* and *F. pubescens*, NADP-ME transcripts are elevated and their amounts increase in parallel with Ala-AT and Asp-AT transcript levels.

To extend the pathways of ammonia recirculation into a rudimentary C$_4$ cycle, a capacity to regenerate malate from pyruvate in the mesophyll is required. As deduced from the transcriptome data, the enzymatic functions required are also already enhanced in the least advanced C$_3$–C$_4$ intermediates, since we observe a low but consistent upregulation of PEPC and PPDK genes in these species

compared to the $C_3$ *Flaverias*. Measurements of radiolabeled $CO_2$ incorporation support the view that a rudimentary $C_4$ cycle is already operating in intermediate *Flaveria* species (*Rumpho et al., 1984*; *Monson et al., 1986*; *Moore et al., 1987*; *Chastain and Chollet, 1989*). *F. chloraefolia* as well as *F. pubescens* incorporate a higher percentage of $^{14}CO_2$ into the $C_4$ compounds malate and aspartate (11.3% and 24.9%) than the $C_3$ species *F. pringlei* and *F. cronquistii* (4.1% and 7.7%) (*Vogan and Sage, 2011*). Thus even the least advanced intermediates analyzed in this study run already a low-level $C_4$ cycle, which assists in recycling the ammonia liberated by GDC in the bundle sheath cells.

The question arises whether amino group transfer initially exclusively happened via amino acid/oxoacid pairs or whether the enzymatic content of $C_3$ plants immediately supported a shuttle that also involved decarboxylation and carboxylation reactions. $C_3$ plants have considerable capacity for the decarboxylation of four-carbon organic acids in their bundle sheath cell (*Hibberd and Quick, 2002*; *Brown et al., 2010*) and measurements of total leaf NAD-ME and NADP-ME activity in $C_3$ plants repeatedly demonstrated basal activities for various $C_3$ species (*Wheeler et al., 2005*; *Aubry et al., 2011*; *Maier et al., 2011*). $C_3$ plants also accumulate high amounts of organic $C_4$ acids like malate or fumarate during the day (*Zell et al., 2010*), which are produced by PEPC, the only enzyme capable of producing $C_4$ acids de novo. It is tempting to hypothesize that plants use a malate/alanine shuttle to recycle parts of the ammonia liberated by glycine decarboxylation from the very beginning of the $C_2$ cycle.

## Elevating the $C_4$ cycle activity in a $C_2$ plant enhances the $CO_2$ fixation capacity

If the $C_4$ cycle is superimposed onto a $C_2$ cycle operating in a $C_3$–$C_4$ intermediate plant, the $C_2$ photosynthesis model predicts a mechanistic interaction between the $C_2$ and $C_4$ cycles (*Figure 5*). When the $C_4$ cycle is running, the photorespiratory ammonia is recirculated from the bundle sheath to the mesophyll cells by moving malate from the mesophyll to the bundle sheath and transferring alanine back to the mesophyll. This malate/alanine cycling leads to a net transport of ammonia from the bundle sheath into the mesophyll cells. In contrast to the other mechanisms of ammonia recirculation described above, the $C_4$ cycle does not only lead to a net transport of ammonia from the bundle sheath to the mesophyll but additionally also to a net transport of $CO_2$ in the opposite direction. Thus $CO_2$ is transferred from the mesophyll to the bundle sheath without increasing the number of transport processes between the cells. By elevating the $CO_2$ concentration in the bundle sheath cells the $C_4$ cycle acts cooperatively with the $C_2$ cycle. The bundle sheath Rubisco would work under a more elevated $CO_2$ concentration and thus operate more effectively compared to a pure $C_2$ plant, leading to an increased biomass production. The $C_4$ cycle thus has a dual beneficial effect: an efficient nitrogen shuttle is combined with a $CO_2$ concentrating pump.

To investigate the possible interaction with regard to biomass, a $C_4$ cycle at the enzyme capacities of $C_3$ plants was allowed and tested for biomass changes (*Figure 5 C*). When the $C_4$ cycle is running with PEPC activities comparable to those found in $C_3$ *Flaveria* species, the model already predicts a gain in biomass production compared to the $C_2$ cycle on its own. Under these conditions, the bulk of photorespiratory ammonia is recycled through a rudimentary $C_4$ cycle limited by the $C_4$ cycle flux capacity. The model predicts that biomass production will be further enhanced with higher activity of the $C_4$ cycle. Consequently, there is permanent positive selection on enhancing the activity of the currently rate limiting enzyme once a $C_4$ cycle is running.

The evolutionary scenario described above is in good agreement with the *Flaveria* transcriptome data. We observe gradual increases in the amounts of $C_4$ transcript with increasing '$C_4$-ness' of the $C_3$–$C_4$ intermediates until the most advanced species *F. brownii.* The abundance of NADP-ME and Ala-AT transcripts increases faster than the transcript abundance of the other core $C_4$ genes like PEPC, PPDK, MDH or Asp-AT. This implies that these evolutionary changes were driven by selection on high bundle sheath decarboxylation capacity, consistent with the idea that the $C_4$ cycle began as an auxiliary pathway to the $C_2$ cycle to recirculate photorespiratory ammonia. Hence, in this early phase, the main purpose of the $C_4$ cycle was to provide the ammonia acceptor pyruvate. The $C_2$ model and its evolutionary implications are consistent with the properties of the $C_3$–$C_4$ intermediate *Flaveria* species including *F. brownii*, which possess mesophyll Rubisco activity and consequently the $C_2$ photosynthetic pathway. The next iteration during $C_4$ evolution in *Flaveria* must have been the restriction of Rubisco activity to the bundle sheath, making the $C_2$ cycle obsolete, as observed for the true $C_4$ *Flaveria* species.

## A scenario for C$_4$ evolution in the genus *Flaveria*—a general blueprint for the evolution of C$_4$ photosynthesis?

The establishment of a photorespiratory CO$_2$ pump, termed C$_2$ photosynthesis, is thought to be an important step in C$_4$ evolution. Recent work has shown how C$_3$ *Flaverias* were preconditioned for the evolution of the C$_2$ pathway and how the C$_2$ cycle was implemented on the molecular level (*Sage et al., 2013*; *Schulze et al., 2013*). Together with the present work, this gives us a detailed picture of what happened in the early and intermediate stages during C$_4$ evolution in *Flaveria*.

We have argued that the establishment of the C$_2$ cycle requires the implementation of at least components of the C$_4$ pathway, if not the whole pathway. This fact might be a partial explanation for the polyphyletic evolution of C$_4$ photosynthesis. Only the C$_2$ cycle has to evolve to set a system on a slippery slope towards C$_4$ photosynthesis. Nature seems to confirm this idea. So far, 66 independent origins of C$_4$ photosynthesis could be identified. In contrast, there are only seven known groups with independent origins of C$_2$ plants and no direct ancestry to C$_4$ species (*Sage et al., 2012*). If one assumes that all recent C$_4$ lineages evolved via C$_2$ intermediates, which appears likely (*Sage et al., 2012*; *Heckmann et al., 2013*; *Williams et al., 2013*), this would mean that the C$_2$ pathway evolved 73 times independently and that over 90% of these C$_2$ plant containing lineages proceeded to C$_4$ photosynthesis. This indicates that the C$_2$ photosynthetic pathway must indeed be a strong enabler of C$_4$ photosynthesis. It will be highly enlightening to analyze these C$_2$ groups without ancestry to C$_4$ species, like *Moricandia*, *Steinchisma* or *Mollugo*, to find out in how far they differ from groups that evolved the C$_4$ pathway and why C$_4$ evolution may have been hampered in these groups.

The close evolutionary interconnection of the C$_2$ and the C$_4$ pathway could be seen as an example of metabolic exaptation (*Barve and Wagner, 2013*). Exaptation or pre-adaptation was defined as an adaptation involving the co-option of traits that originally evolved for a different purpose (*Gould and Vrba, 1982*). While both C$_2$ and C$_4$ act as carbon shuttles to the bundle sheath cells, the two systems achieve this goal through different biochemical processes. In particular, the amino acid shuttle in the C$_2$ system evolved to transport nitrogen, and its later use in C$_4$ photosynthesis to shuttle carbon thus represents a molecular exaptation. Our findings therefore corroborate the general idea that the evolution of complex traits may be accelerated through exaptations (*Darwin, 1872*; *Gould and Vrba, 1982*; *Barve and Wagner, 2013*).

We do not know if the scenario on the early and intermediate stages of evolution described above is limited to the genus *Flaveria* or if it is valid for C$_4$ evolution in general. Our prediction of the C$_2$ pathway being a strong facilitator of C$_4$ evolution should apply to all C$_4$ origins, as the integrated model is not specific to *Flaveria*.

## Materials and methods

### Plant material

*F. pringlei*, *F. robusta*, *F. chloraefolia*, *F. pubescens*, *F. anomala*, *F. ramosissima*, *F. brownii*, *F. bidentis and F. trinervia* plants were grown in the green house at University of Duesseldorf side-by-side and harvested at four different points of time over the year. The plants were grown in 17-cm pots on soil (C-400 with Cocopor [Stender Erden, Schermbeck Germany] fertilized with 3 g/l Osmocote exact standard 3 to 4 M [Marysville, USA]) with additional light for 16 hr per day until 50 to 60 cm height and before the onset of flowering.

Plants for experiment one were harvested in September, for experiment two in June, for experiment three in October and for experiment four in April. The plant material was immediately frozen in liquid nitrogen, stored at −80°C and used for the following analyses.

### RNA isolation, transcriptome sequencing and analysis

Total RNA was isolated from the second and fourth leaves according to (*Westhoff et al., 1991*) followed by a DNAse treatment. After phenol/chloroform extraction and precipitation with NaAc and isopropyl alcohol the RNA was dissolved in H$_2$O. The RNA quality was tested with the Agilent 2100 bioanalyzer. 1 µg of total RNA was used for cDNA library generation, which was accomplished with the TruSeq RNA Sample Preparation Kit (Illumina Inc., San Diego, USA) via the Low-Throughput Protocol (TruSeq RNA Sample Preparation Guide, Illumina Proprietary Catalog # RS-930-2001, Part # 15008136 Rev. A, November 2010). Clusters were generated with the TruSeq SR Cluster Kit v2 according to the

Reagent Preparation Guide with the Illumina cBot device. The single read sequencing was performed with the Illumina HiSeq2000.

Sequences of transcripts from genes involved in photorespiraton, C₄ photosynthesis and refixation and recirculation of photorespiratory ammonia were identified among de novo assembled transcripts of *F. robusta.* De novo assembly was performed with either CLC Genomics Workbench (CLC-Bio, Aarhus, Denmark) or the Velvet/Oases software package (*Schulz et al., 2012*) using *F. robusta* 454 (*Gowik et al., 2011*) and Illumina reads (this study).

After quality control and processing, Illumina reads were aligned to the *F. robusta* transcript sequences with the CLC Genomics Workbench using standard parameters. Read mapping against a minimal set of coding sequences (*Brautigam et al., 2011*) of the TAIR 9 release of the *Arabidopsis thaliana* genome (http://www.Arabidopsis.org/) was performed using BLAT (*Kent, 2002*) as described in (*Gowik et al., 2011*).

The MEV software package (http://www.tm4.org/mev.html) was used for plotting heat maps, hierarchical clustering and principal component analysis.

## Protein isolation and quantification

Total proteins were isolated from plant material harvested together with the material for RNA isolation according to *Shen et al. (2007)* and quantified using the RC-DC protocol (Bio-Rad Laboratories, Hercules, USA). 30 µg of total protein was electrophoresed on polyacrylamide-SDS gels (*Schägger and von Jagow, 1987*) and electrophoretically transferred to nitrocellulose membranes (Protran BA85, 0.45 µm; Schleicher & Schuell, Dassel, Germany) for 1 hr with 0.8 mA per cm². Specific primary antibodies were raised against conserved *Flaveria* peptides (Agrisera Vännäs, Sweden). For the detection of specific proteins the nitrocellulose membranes were incubated with the primary antibodies and a Horseradish peroxidase-conjugated secondary antibody (Sigma-Aldrich, St. Louis, USA). An enhanced chemiluminescent Horseradish peroxide substrate was added and signals were recorded using a Fuji LAS-4000 mini CCD camera system. The signals were quantified with the Multi Gage analysis software (Fujifilm, Tokyo, Japan). As loading control a gel was stained for 45 min with 0.25% Coomassie blue, 50% methanol, 7% acetic acid, and destained in 50% methanol, 7% acetic acid.

## Coupling a mechanistic model with a genome-scale metabolic reconstruction

In order to model the metabolic integration of C₂ and C₄ cycle in the context of leaf metabolism, we conducted Flux Balance Analysis (FBA) based on a genome-scale metabolic reconstruction of C₄ metabolism, C4GEM (*Dal'Molin et al., 2010*). This reconstruction contains a complex biomass reaction including carbohydrates, cell wall components, amino acids and nucleotides (*Dal'Molin et al., 2010*).

FBA is a powerful tool to understand the adaptation of metabolism on a genomic scale. Since metabolite concentrations are not modeled explicitly, fluxes related to carbon concentration mechanisms (CCMs) cannot be captured by this constraint-based approach alone. To account for this issue, we coupled the FBA model with a mechanistic model of C₃–C₄ photosynthesis (*von Caemmerer, 2000*; *Heckmann et al., 2013*).

C4GEM representing NADP-ME types was provided by the authors and FBA was conducted using this model:

$$\text{Maximize } c^T v$$

subject to $Sv = 0$.

$$v_{min,i} \leq v_i \leq v_{max,i}$$

where $c$ is the vector of coefficients in the objective function, here the leaf biomass production. $v$ is the vector of fluxes through the network reactions, $S$ is the stoichiometric matrix of the metabolic network, and $v_{min}$ and $v_{max}$ represent constraints on the respective fluxes.

In order to test hypotheses concerning nitrogen metabolism in C₃–C₄ intermediate plants, $S$ had to be modified. The plasmosdesmatal transport reactions in the original C4GEM model include malate, pyruvate, 3-phosphoglycerate, trioses, phosphates, sucrose, aspartate, alanine, phosphoenolpyruvate, CO₂, and O₂. Reactions were added to $S$ in order to include transport of serine, glycine, glutamate, glutamine, asparagine, threonine, 2-oxoglutarate and water over the mesophyll/bundle sheath interface. Furthermore, the lack of photosystem II in the bundle sheath of certain C₄ plants does not

hold in our scenario (**Nakamura et al., 2013**) and we added a reaction for linear electron transport to the bundle sheath. C4GEM does not contain a reaction for a plastidal NADP-dependent malate dehydrogenase in the bundle sheath; we added this reaction to S.

In addition to the stoichiometric matrix S, the constraints used in C4GEM were modified:

The original constraint on leaf sucrose production was changed to result in an output ratio of sucrose to amino acids of about 5 (**Riens et al., 1991**). Fixed constraints on production of starch and fatty acids are not appropriate in the coupled framework. Since we are not aware of data that explains how these fluxes scale with net $CO_2$ assimilation rate, the constraints were removed from the model. Reactions belonging to the GS/GOGAT system were assumed to be irreversible. Nitrogen is available in the form of nitrate as opposed to $NH_3$ in the original model. Since there is no evidence suggesting mesophyll specificity of PEPC in intermediate *Flaveria* species, we unconstrained PEPC flux in the bundle sheath.

To couple the genome-scale FBA model with the mechanistic model of carbon fixation, the following reactions were constrained using the values predicted by the mechanistic model: net $CO_2$ uptake, Rubisco carboxylation and oxygenation in mesophyll and bundle sheath, $CO_2$ leakage from the bundle sheath, PEPC activity in the mesophyll, activity of NADP-ME in the bundle sheath, plasmodesmatal flux of glycine and serine and decarboxylation by the GDC complex. The lower bound on glycine diffusion ($V_{min,Gly}$), serine diffusion ($V_{min,Ser}$), and GDC reaction ($V_{min,GDC}$) can be obtained from the rate of Rubisco oxygenation in the mesophyll ($V_{om}$) and the fraction of photorespiratory $CO_2$ in the bundle sheath derived from mesophyll oxygenations ($\xi$):

$$v_{min,Gly} = \xi V_{om}, \; v_{min,Ser} = 0.5\xi V_{om}, \; v_{min,GDC} = 0.5\xi V_{om}$$

The mechanistic model was parameterized to the $C_3$ state as given in **Heckmann et al. (2013)**, with the exception of the parameter $\xi$, which was set to a value of 0.98 (*i.e.*, the majority of GDC activity was restricted to the bundle sheath. Derivation from transcriptome data is given in **Heckmann et al. (2013)**). These constraints on the reactions of the photorespiratory pump are necessary to adequately predict $C_2$ photosynthesis because of the inability of FBA alone to model CCMs (see discussion above).

In the FBA part of the model, a minimization of total flux (MTF) analysis was conducted in order to narrow down the space of optimal solutions:

$$\text{Minimize} \sum_{i=1}^{n} w_i |v_i|$$

subject to: $Sv = 0$.

$$v_{min,i} \leq v_i \leq v_{max,i}$$

$$c^T v = c^T v_{FBA}$$

where $v_{FBA}$ is the flux distribution of the FBA optimization described above. $w$ denotes a vector of weights, where plasmodesmatal flux received a higher weighting factor (1.1 for plasmodesmatal exchange, 1 for the remaining reactions). This method implements a simple minimization of protein costs for a given optimal biomass production. The higher weights on plasmodesmatal fluxes account for the trade-off between $CO_2$ containment in the bundle sheath and metabolite diffusion between the cells. Since this trade-off is difficult to quantify, we conducted a sensitivity analysis by varying the weight on plasmodesmatal transport reactions.

In order to investigate the possible range that fluxes can take while yielding an optimal solution, flux variability analysis was conducted:

For each $v_i$:

Maximize or Minimize $v_i$.

Subject to: $Sv = 0$.

$$v_{min,i} \leq v_i \leq v_{max,i}$$

$$c^T v = c^T v_{FBA}$$

$$\sum_{i=1}^{n} w_i|v_i| = s_{opt}$$

Where $s_{opt}$ is the minimum for the weighted sum of absolute flux found in the MTF optimization.

All simulations were conducted in the R environment for statistical computing (*R Core Team, 2013*) using the sybil library (*Gelius-Dietrich et al., 2013*).

## Accession numbers

The read data have been submitted to the National Center for Biotechnology Information Short Read Archive under accession numbers SRP036880 (*F. bidentis*), SRP036881 (*F. anomala*), SRP036883 (*F. brownii*), SRP036884 (*F. chloraefolia*), SRP036885 (*F. pringlei*), SRP037526 (*F. pubescens*), SRP037527 (*F. ramosissima*), SRP037528 (*F. robusta*) and SRP037529 (*F. trinervia*).

## Acknowledgements

We thank the 'Genomics and Transcriptomics laboratory' of the 'Biologisch-Medizinischen Forschungszentrum' (BMFZ) at the Heinrich-Heine-University of Duesseldorf (Germany) for technical support and conducting the Illumina sequencing.

## Additional information

### Funding

| Funder | Grant reference number | Author |
|---|---|---|
| Deutsche Forschungsgemeinschaft | FOR 1186 | Andreas PM Weber, Peter Westhoff |
| Deutsche Forschungsgemeinschaft | IRTG 1525 | David Heckmann |
| Deutsche Forschungsgemeinschaft | CRC 680 | Martin J Lercher |
| Deutsche Forschungsgemeinschaft | EXC 1028 | Martin J Lercher, Andreas PM Weber, Peter Westhoff |
| European Union | EU Project 3to4 | Andreas PM Weber, Peter Westhoff |

The funders had no role in study design, data collection and interpretation, or the decision to submit the work for publication.

### Author contributions

JM, Conception and design, Acquisition of data, Analysis and interpretation of data, Drafting or revising the article; DH, In silico modeling, Conception and design, Analysis and interpretation of data, Drafting or revising the article; AB, UG, Conception and design, Analysis and interpretation of data, Drafting or revising the article; MJL, APMW, PW, Conception and design, Drafting or revising the article

## Additional files

### Major datasets

The following datasets were generated:

| Author(s) | Year | Dataset title | Dataset ID and/or URL | Database, license, and accessibility information |
|---|---|---|---|---|
| Mallmann J, Heckmann D, Bräutigam A, Lercher MJ, Weber APM, Westhoff P, Gowik U | 2014 | Flaveria bidentis leaf transcriptomes | http://www.ncbi.nlm.nih.gov/sra/?term=SRP036880 | Publicly available at NCBI Short Read Archive (http://www.ncbi.nlm.nih.gov/sra). |

| | | | | |
|---|---|---|---|---|
| Mallmann J, Heckmann D, Bräutigam A, Lercher MJ, Weber APM, Westhoff P, Gowik U | 2014 | Flaveria anomala leaf transcriptomes | http://www.ncbi.nlm.nih.gov/sra/?term=SRP036881 | Publicly available at NCBI Short Read Archive (http://www.ncbi.nlm.nih.gov/sra). |
| Mallmann J, Heckmann D, Bräutigam A, Lercher MJ, Weber APM, Westhoff P, Gowik U | 2014 | Flaveria brownii leaf transcriptomes | http://www.ncbi.nlm.nih.gov/sra/?term=SRP036883 | Publicly available at NCBI Short Read Archive (http://www.ncbi.nlm.nih.gov/sra). |
| Mallmann J, Heckmann D, Bräutigam A, Lercher MJ, Weber APM, Westhoff P, Gowik U | 2014 | Flaveria chloreafolia leaf transcriptomes | http://www.ncbi.nlm.nih.gov/sra/?term=SRP036884 | Publicly available at NCBI Short Read Archive (http://www.ncbi.nlm.nih.gov/sra). |
| Mallmann J, Heckmann D, Bräutigam A, Lercher MJ, Weber APM, Westhoff P, Gowik U | 2014 | Flaveria pringlei leaf transcriptomes | http://www.ncbi.nlm.nih.gov/sra/?term=SRP036885 | Publicly available at NCBI Short Read Archive (http://www.ncbi.nlm.nih.gov/sra). |
| Mallmann J, Heckmann D, Bräutigam A, Lercher MJ, Weber APM, Westhoff P, Gowik U | 2014 | Flaveria pubescens leaf transcriptomes | http://www.ncbi.nlm.nih.gov/sra/?term=SRP037526 | Publicly available at NCBI Short Read Archive (http://www.ncbi.nlm.nih.gov/sra). |
| Mallmann J, Heckmann D, Bräutigam A, Lercher MJ, Weber APM, Westhoff P, Gowik U | 2014 | Flaveria ramosissima leaf transcriptomes | http://www.ncbi.nlm.nih.gov/sra/?term=SRP037527 | Publicly available at NCBI Short Read Archive (http://www.ncbi.nlm.nih.gov/sra). |
| Mallmann J, Heckmann D, Bräutigam A, Lercher MJ, Weber APM, Westhoff P, Gowik U | 2014 | Flaveria robusta leaf transcriptomes | http://www.ncbi.nlm.nih.gov/sra/?term=SRP037528 | Publicly available at NCBI Short Read Archive (http://www.ncbi.nlm.nih.gov/sra). |
| Mallmann J, Heckmann D, Bräutigam A, Lercher MJ, Weber APM, Westhoff P, Gowik U | 2014 | Flaveria trinervia leaf transcriptomes | http://www.ncbi.nlm.nih.gov/sra/?term=SRP037529 | Publicly available at NCBI Short Read Archive (http://www.ncbi.nlm.nih.gov/sra). |
| Mallmann J, Heckmann D, Bräutigam A, Lercher MJ, Weber APM, Westhoff P, Gowik U | 2014 | Data from: the role of photorespiration during the evolution of C4 photosynthesis in the genus Flaveria | http://dx.doi.org/10.5061/dryad.q827h | Available at Dryad Digital Repository under a CC0 Public Domain Dedication. |

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
