## [Decision Letter]

Thank you for sending your work entitled “The role of photorespiration during the evolution of C_4_photosynthesis in the genus *Flaveria*” for consideration at *eLife*. Your article has been favorably evaluated by a Senior editor (Detlef Weigel) and 3 reviewers, one of whom, Chris Pires, has agreed to reveal his identity.

The Senior editor and the reviewers discussed their comments before we reached this decision, and the Senior editor has assembled the following comments to help you prepare a revised submission.

This is a genuinely exciting and provocative study; if the results hold up, they are extremely important for the field, and represent a major advance in our understanding of C_4_ evolution.

The hypothesis tested is that the C_2_ photosynthesis pump is an evolutionary precursor in the shift from C_3_ to C_4_ photosynthesis (which has happened multiple times). Across nine Flaveria species that includes C_3_, C_4_ and intermediate species taxa with intermediate C_3_ and C_4_ photosynthesis, transcriptome and protein data were collected and analyzed in a phylogenetic context. A global genome-wide metabolic model and flux balance analysis (FBA) was used to integrate models with transcript abundance. The coordinated changes in expression of all main photorespiratory enzymes strongly suggest an evolutionary scenario for the origin of C_4_ photosynthesis. Very little attention has been paid in the past to nitrogen effects of C_2_, yet the problem is obvious, and if the C_4_ cycle is the most direct way to rebalance N within the leaf tissue, then this route provides an immediate solution to a very important and poorly understood transition in the trajectory. It implies that C_2_ crosses a new threshold of C_4_ accessibility, and that C_4_ is all but an inevitable outcome. This also can partially explain the relative rarity of known C_2_ phenotypes – this may be a part of phenotypic space with a low 'persistence time' – an area that lineages rarely evolve into and very quickly evolve out of.

Key to your argument is whether or not the 're-tooling' of the C4GEM model to accommodate a C_2_ system was done properly. I'd suggest getting an additional reviewer with expertise in metabolic modeling for that. The analysis done is fairly standard, but relies on several assumptions and choices that do not seem to be justified in the present version of the manuscript. In addition, robustness checks, particularly with respect to the uniqueness of the solutions, need to be obtained, analyzed, and subsequently interpreted.

Details:

1) How could the selected 11 fluxes be fixed from the mechanistic model? The cited mechanistic models (68; 28) aim at predicting CO_2_ assimilation rate given a set of parameter values (e.g., Michaelis-Menten constants) as well as input values for partial pressure of CO_2_ and O_2_. The cited mechanistic model does not include the plasmodesmatal flux of glycine and serine or the decarboxylation of the GDC. Moreover, the CO_2_ uptake rate is assumed (not predicted by the model). These points need to be further elaborated on.

2) You state that you maximized biomass production rate. However, you do not comment on why fixed constraints on production of starch and fatty acids were removed; this should be supported by literature or clear reasons should be provided. The remaining model modifications seem plausible.

3) The second objective applied is the minimization of the sum of absolute fluxes including transport processes (at fixed optimal biomass production rate). In doing so, higher weight was assigned to plasmodesmatal fluxes (expecting that they receive lower flux). However, as shown in the Methods section, the weight for the plasmodesmatal fluxes is seemingly arbitrarily chosen to be 1.1. What effect would the increase of this value to, say, 2 or 5 have on the drawn conclusions from this study? Either clear reasoning for selection of the value 1.1 should be provided (the mentioned trade-off does justify the selection of the value) or additional tests should be carried out to explore the consequences of other (biologically reasonable) weights for these reactions.

4) You state that this framework allows the prediction of detailed flux distributions that are biologically realistic. In the Methods section, you also indicate that the combination of several objectives was used to reduce the size of the space of optimal solutions. However, you do not provide a statement on whether or not the resulting distributions are indeed unique (e.g., the one provided in Figure 4). If the distribution for a fixed biomass production and a fixed sum of absolute flux values is not unique, the provided findings and interpretation are just one of many available alternatives! This point should be carefully examined (e.g., by Flux variability analysis); this analysis must be coupled with the issues raised in point 3, above.

5) The alternative solutions with a larger number of reactions are presented without providing how much they differ in terms of the total absolute values of flux from that of the preferred glutamate/2-oxoglutarate solution. The increase in the total absolute plasmodesmatal flux, evident in Figure 4, should be discussed with respect to biological implications.

6) The statement “In terms of biomass production rate, this shuttle represents another alternative optimal solution” in the Results section, is confusing and may be misleading. Is this an alternative solution for the same total absolute values of flux as the glutamate/2-oxoglutarate solution? If not, this sentence can be dropped since it was already indicated that alternatives were selected to “retain the same biomass output”.

7) In the Results section, what do you mean by “In a restrictive scenario, all nitrogen containing compounds were excluded, except for glycine and serene”? What does that do to the reactions that use these metabolites? Are they also removed?

8) The prediction of the model discussed in the Results section should be investigated with respect to the issues raised in points 3 – 5, above.

[Editors' note: further clarifications were requested prior to acceptance, as described below.]

Thank you for resubmitting your work entitled “The role of photorespiration during the evolution of C_4_ photosynthesis in the genus *Flaveria*” for further consideration at eLife. Your revised article has been favorably evaluated by Detlef Weigel (Senior editor) and a member of the Board of Reviewing Editors. The manuscript has been improved but there are some remaining issues that need to be addressed before acceptance, as outlined below:

The authors have carefully considered the suggestions provided about the earlier version of the manuscript and have: (1) offered detailed explanations of how the results of the mechanistic model were used to constrain the larger stoichiometric model as well as (2) conducted flux variability analysis for the different alternative scenarios to support the robustness of the predictions.

The only remaining concern deals with the details on lower/upper boundary selection for the diffusion fluxes and their effects on the predictions. Additional clarifications (and/or analyses) may further strengthen the predictions: The lower bounds on the diffusion of gylcine, serine, and NADPME reaction are fixed based on oxygenation rate and the fraction of photorespiratory CO_2_ in the bundle sheath derived from mesophyll oxygenations.

1) Please, indicate the value for the latter parameter (i.e., the fraction of photorespiratory CO_2_ in the bundle sheath derived from mesophyll oxygenations).

2) To thoroughly examine the viability of the modeling strategy, it is also necessary to determine the variability of model predictions without specifying tight lower (and upper) boundaries (as several fluxes are specified together with two optimization criteria), which may a priori fix the solution. What is the variability of the solutions with default lower/upper boundaries (say, of 0/1000 (irreversible case)) for the diffusion fluxes?

This additional analysis can be readily conducted and included in the manuscript.

---

## [Author Response]

*1) How could the selected 11 fluxes be fixed from the mechanistic model? The cited mechanistic models (*[68]*;*
[28]*) aim at predicting CO*_*2*_
*assimilation rate given a set of parameter values (e.g., Michaelis-Menten constants) as well as input values for partial pressure of CO*_*2*_
*and O*_*2*_*. The cited mechanistic model does not include the plasmodesmatal flux of glycine and serine or the decarboxylation of the GDC. Moreover, the CO*_*2*_
*uptake rate is assumed (not predicted by the model). These points need to be further elaborated on*.

We apologize for not explaining these points in more detail. The mentioned fluxes are indeed not described explicitly, but are described by the model implicitly. We expanded the Method section to show how these were included. The net CO_2_ uptake rate is predicted by the mechanistic model and can be readily used as a constraint in the flux-balance analysis.

*2) You state that you maximized biomass production rate. However, you do not comment on why fixed constraints on production of starch and fatty acids were removed; this should be supported by literature or clear reasons should be provided. The remaining model modifications seem plausible*.

The original C4GEM model does not contain starch and fatty acid production as part of the biomass function. Instead, the fluxes are fixed by constraints. Since we are not aware of data that explains how these fluxes scale with CO_2_ assimilation rate, we removed the respective constraints from the model. This rationale is now described in more detail in the Methods section.

*3) The second objective applied is the minimization of the sum of absolute fluxes including transport processes (at fixed optimal biomass production rate). In doing so, higher weight was assigned to plasmodesmatal fluxes (expecting that they receive lower flux). However, as shown in the Methods section, the weight for the plasmodesmatal fluxes is seemingly arbitrarily chosen to be 1.1. What effect would the increase of this value to, say, 2 or 5 have on the drawn conclusions from this study? Either clear reasoning for selection of the value 1.1 should be provided (the mentioned trade-off does justify the selection of the value) or additional tests should be carried out to explore the consequences of other (biologically reasonable) weights for these reactions*.

Although the trade-off between metabolite transport and CO_2_ containment suggests additional weight on plasmodesmatal fluxes, we have to admit that an exact value cannot be inferred from available data. To account for this issue, we conducted a sensitivity analysis against the weight on plasmodesmatal flux. The results presented in Figure 4 are generally stable against variation in weights. Interestingly, the shuttles αKG/Glu and the Mal/Asp shuttle are extended to use nitrogen rich compounds (Gln and Asn respectively) when very high weights are applied. We included these findings in the Results section. The resulting flux over plasmodesmata including flux variability (see point 4) can be found as part of the Supplementary Material.

*4) You state that this framework allows the prediction of detailed flux distributions that are biologically realistic. In the Methods section, you also indicate that the combination of several objectives was used to reduce the size of the space of optimal solutions. However, you do not provide a statement on whether or not the resulting distributions are indeed unique (e.g., the one provided in*
Figure 4*). If the distribution for a fixed biomass production and a fixed sum of absolute flux values is not unique, the provided findings and interpretation are just one of many available alternatives! This point should be carefully examined (e.g., by Flux variability analysis); this analysis must be coupled with the issues raised in point 3, above*.

Our rationale behind assuming a small solution space was the application of the two optimality criteria; but we agree with you entirely that this should be shown more rigorously. We thus implemented a flux variability analysis and added the results to the fluxes shown in Figures 4 and 5. The analysis shows that the space of optimal solutions is small and only minuscule variability in the fluxes of the shuttles is possible. The algorithm is described in the Methods and the Results section of the main text.

*5) The alternative solutions with a larger number of reactions are presented without providing how much they differ in terms of the total absolute values of flux from that of the preferred glutamate/2-oxoglutarate solution. The increase in the total absolute plasmodesmatal flux, evident in*
Figure 4*, should be discussed with respect to biological implications*.

The sum of absolute flux over plasmodesmata was added to the legend of Figures 4 and 5.

*6) The statement “In terms of biomass production rate, this shuttle represents another alternative optimal solution” in the Results section, is confusing and may be misleading. Is this an alternative solution for the same total absolute values of flux as the glutamate/2-oxoglutarate solution? If not, this sentence can be dropped since it was already indicated that alternatives were selected to “retain the same biomass output”*.

We agree with you that this point does not need further emphasis and removed the sentence.

7) In the Results section, what do you mean by “In a restrictive scenario, all nitrogen containing compounds were excluded, except for glycine and serene”? What does that do to the reactions that use these metabolites? Are they also removed?

We apologize for not stating that we exclusively excluded plasmodesmatal nitrogen compounds and changed the sentence to make this point clear to the reader.

*8) The prediction of the model discussed in the Results section should be investigated with respect to the issues raised in points 3 – 5, above*.

We now mention the alternative solution of the flux distribution shown in Figure 5 and the trade-off between metabolite diffusion and gas containment in the Results section.

[Editors' note: further clarifications were requested prior to acceptance, as described below.]

*1) Please, indicate the value for the latter parameter (i.e., the fraction of photorespiratory CO*_*2*_
*in the bundle sheath derived from mesophyll oxygenations)*.

We indicate the fraction of photorespiratory CO_2_ in the bundle sheath derived from mesophyll oxygenation (0.98 – the derivation from transcriptome data is given in [28]) now in the Methods section of the manuscript.

2) To thoroughly examine the viability of the modeling strategy, it is also necessary to determine the variability of model predictions without specifying tight lower (and upper) boundaries (as several fluxes are specified together with two optimization criteria), which may a priori fix the solution. What is the variability of the solutions with default lower/upper boundaries (say, of 0/1000 (irreversible case)) for the diffusion fluxes?

The rationale behind our modeling approach was to incorporate current mechanistic knowledge about C_2_ photosynthesis in *Flaveria* in such a way that we can answer the question: Given our current understanding of C_2_ photosynthesis, what is the implementation of this pathway in the plant leaf that optimizes fitness-relevant criteria?

In our opinion, the most reasonable approach to answering this question was the description of the current conceptual model of C_2_ photosynthesis (that includes diffusion of glycine and serine) through mechanistic equations and to yield lower bounds for the respective reactions in the stoichiometric model. We understand the reviewers’ question as an equivalent of asking:

Why is it necessary to yield lower bounds for the fluxes of the photorespiratory pump from the mechanistic model? If C_2_ photosynthesis increases biomass yield, shouldn’t FBA be able to predict such a cycle without the need for further constraints?

As stated in the Methods section of the manuscript Flux Balance Analysis is a powerful tool but it cannot model metabolite concentrations explicitly. Therefore fluxes related to concentration-dependent processes, like carbon concentration mechanisms, cannot be captured by this constraint-based approach. To account for this issue we used values, predicted by the mechanistic model of C_3_-C_4_ photosynthesis, for constraining certain fluxes within the FBA model to integrate C_2_ photosynthesis by creating flux through this pathway. The mechanistic model explicitly accounts for concentrations of CO_2_ and O_2_ in the two cell types and predicts the rates of rubisco carboxylation and oxygenation that result from these concentrations. The fluxes mentioned by the reviewer: diffusion of glycine, serine and the GDC reaction belong to these fluxes that have to be constrained to ensure flux through the C_2_ cycle. When we remove these constraints the model will not predict C_2_ photosynthetic activity anymore.

We checked this and indeed fluxes through the C_2_ photosynthetic cycle, but also the predicted fluxes to recirculate ammonia from bundle sheath to mesophyll cells, disappear when we remove constraints on the plasmodesmatal diffusion of glycine, serine, and CO_2_ and the GDC reaction. We added a sentence to the Methods section to clarify this point.